# Dual Alignment Framework for Few-shot Learning with Inter-Set and Intra-Set Shifts

**Siyang Jiang**[1], **Rui Fang**[2], **Hsi-Wen Chen**[2], **Wei Ding**[2]
**Guolaing Xing**[✉,1], **Ming-Syan Chen**[✉,2]
[1]The Chinese University of Hong Kong
[2]National Taiwan University

## Abstract

Few-shot learning (FSL) aims to classify unseen examples (query set) into labeled data (support set) through low-dimensional embeddings. However, the diversity and unpredictability of environments and capture devices make FSL more challenging in real-world applications. In this paper, we propose *Dual Support Query Shift (DSQS)*, a novel challenge in FSL that integrates two key issues: inter-set shifts (between support and query sets) and intra-set shifts (within each set), which significantly hinder model performance. To tackle these challenges, we introduce a *DUal ALignment framework (DUAL)*, whose core insight is that clean features can improve optimal transportation (OT) alignment. Firstly, DUAL leverages a robust embedding function enhanced by a repairer network trained with perturbed and adversarially generated "hard" examples to obtain clean features. Additionally, it incorporates a two-stage OT approach with a negative entropy regularizer, which aligns support set instances, minimizes intra-class distances, and uses query data as anchor nodes to achieve effective distribution alignment. We provide a theoretical bound of DUAL and experimental results on three image datasets, compared against 10 state-of-the-art baselines, showing that DUAL achieves a remarkable average performance improvement of 25.66%. Our code is available at https://github.com/siyang-jiang/DUAL.

## 1 Introduction

Few-shot learning (FSL) addresses the challenge of limited labeled data by extracting features and leveraging the similarity between support and query sets, rather than training a separate classifier for each class. This characteristic makes FSL suitable for tasks with scarce data and unseen scenarios, as exemplified by methods such as MatchingNet [45], which assigns a query example the label of its most similar counterpart in the support set. Conventional studies on FSL often focus on cross-domain settings [37, 52], where distribution shifts occur between training and testing data [41, 26]. To mitigate this issue, previous approaches have enhanced robustness through data augmentation [49, 50] or adversarial training [18, 53].

However, these methods assume that each support or query set is internally consistent, i.e., they share the same domain during testing. In practice, *Support-Query Shift (SQS)* [3] frequently arises due to differences in environments (e.g., foggy vs. high-luminance scenes) or capture devices (e.g., mobile phones vs. SLR cameras), leading to misclassification. To address SQS, Bennequin et al. [3] first employed optimal transportation (OT) [7] to align embeddings into a shared latent space. More recently, Jian et al. [21] introduced a noise-aware data augmentation scheme to alleviate distribution misalignment, while Aimen et al. [1] highlighted the growing distribution mismatch between support and query sets during testing.

39th Conference on Neural Information Processing Systems (NeurIPS 2025).

Yet prior studies on SQS have primarily addressed *inter-set shifts*, i.e., differences between support and query sets, while often overlooking *intra-set shifts*, where instances within the same set experience distinct disturbances. These intra-set shifts further complicate the problem by blurring decision boundaries. We term this overlooked issue *Dual Support-Query Shift (DSQS)*, which encompasses both inter-set and intra-set shifts during meta-testing. Intra-set variations can be as significant as inter-set shifts, posing substantial challenges for existing SQS mitigation methods.

As shown in Fig. 1, panel (a) illustrates that conventional FSL, with no shift between support and query sets, exhibits clear decision boundaries. These boundaries, however, become blurred under either inter-set shifts (b) or intra-set shifts (c). In the inter-set case, samples of the same class in the support and query sets may still cluster, yet the comparison module fails to classify query instances correctly because the two sets lie in different domains. In the intra-set case, instances of the same class within a set are scattered across domains, preventing clustering. Consequently, even if a query is classified to a nearby support instance, it may not belong to the same class. When both shifts occur simultaneously, as shown in (d), the boundaries blur even further, severely reducing generalization and leading to poor inference performance under DSQS.

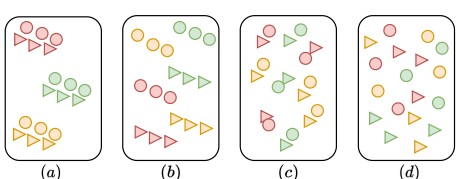

Figure 1: An illustrative example of Support-Query Shifts (SQS), where colors indicate classes, circles denote support samples, and triangles denote query samples: (a) no shift, (b) inter-set shift, (c) intra-set shift, and (d) dual shifts (DSQS).

To address the DSQS problem, we propose the *DUal ALignment Framework (DUAL)*, designed to mitigate the adverse effects of two types of distribution shifts: inter-set shifts (between support and query sets) and intra-set shifts (within each set). Based on our theoretical analysis, DUAL alleviates the challenges in optimal transportation by combining a robust embedding function with a pixel-level repairer to obtain clean features. The repairer, trained on predefined shifts that simulate query distortions, rectifies them by minimizing the feature-space distance between original and repaired data, thereby counteracting both inter-set and intra-set shifts.

In addition, the robust embedding function is trained using a generator that adversarially produces perturbed "hard" samples that are less similar in the embedding space yet still correctly classified. DUAL then employs a dual-regularized optimal transport approach, which identifies class-oriented anchors within the support set by minimizing intra-class distances and aligns the distribution of other instances to these anchors using optimal transport with a negative entropy regularizer. Additional query samples are incorporated as anchors to enhance the robustness of the transportation plan.

The main contributions of this work are summarized as follows.

- We propose the **Dual Support-Query Shift (DSQS)** challenge, which investigates the inter-set and intra-set shift problems in FSL. We theoretically show that both types of shifts can misguide the domain alignment process under optimal transportation.

- To address DSQS, we introduce the **DUal ALignment Framework (DUAL)**, which leverages a repairer together with a robust embedding function adversarially trained by a generator to obtain clean features. These features are then used to align support and query distributions through dual-regularized optimal transportation.

- We provide both theoretical and empirical analyses of DUAL. In particular, we theoretically characterize its behavior, and extensive experiments demonstrate that DUAL consistently outperforms 10 state-of-the-art methods, achieving an average improvement of 25.66% across three benchmark datasets.

## 2 Related Work

**Support-Query Shift in Few-shot Learning**    Conventional few-shot learning (FSL) methods can be broadly categorized into three groups: hallucination-based, optimization-based, and metric-based approaches [36]. Hallucination- and optimization-based methods typically aim to obtain a strong initial model that can quickly adapt to new tasks with minimal updates [24, 33, 47, 48]. Our work is more closely related to metric-based FSL, which focuses on learning a similarity-based classi-

fier [45, 42, 17]. Representative methods include MatchingNet [45], which uses pairwise metrics, and ProtoNet [42], which relies on class-wise metrics to assign labels to query samples based on their proximity to support set representations. More recently, the *support-query shift* (SQS) setting has emerged, where the support and query sets are drawn from different domains. To address this challenge, TP [3] leverages optimal transport (OT) to align the support and query distributions. However, image perturbations can distort the transport plan and lead to suboptimal alignment. To mitigate this issue, PGADA [21] integrates a regularized OT framework with an adversarial generator to produce challenging examples for self-supervised adaptation. Similarly, AQP [1] enhances model robustness by generating challenging instances but relies on a projection method rather than adversarial generation.

**Robust Few-shot Learning** Robust few-shot learning aims to defend against adversarial samples by developing robust embedding functions [12, 35, 28]. One research direction focuses on the use of adversarial queries. AQ [12] employs adversarial queries to enhance model robustness, while LCAT [29], a meta-learning-based method, achieves comparable performance to AQ with reduced training time. In addition, Dong et al. [10] propose a non-meta-learning method that learns a robust embedding function and applies a post-processing feature purifier to reduce computational overhead further. SimpleFS [43] trains a robust network on base samples and classifies new samples by assigning them to the nearest base-category centroids in the feature space. Another line of research leverages high-frequency spectrum information or self-distillation, both of which are sensitive to adversarial perturbations in those regions [46, 27, 35]. For instance, LFI [27] shows that publicly available robust models prefer the low-frequency spectrum, thereby avoiding other adversarial perturbations. SSL-ProtoNet [28] employs self-distillation to build robust classifiers, while RAS [35] uses adversarial self-distillation to achieve robustness without explicitly using adversarial samples.

## 3 Preliminary

**Few-shot Learning.** Conventional FSL methods can be broadly categorized into three groups: hallucination-based, optimization-based, and metric-based approaches. In metric-based FSL, a support set $\mathcal{S} = \bigcup_{c \in \mathcal{C}} \mathcal{S}^c$ consists of $\mathcal{C}$ classes, where each class $c$ contains $|\mathcal{S}^c|$ labeled instances. The objective of FSL is to correctly assign each element of the query set $\mathcal{Q} = \bigcup_{c \in \mathcal{C}} \mathcal{Q}^c$ to one of these $\mathcal{C}$ classes.

Let $\phi$ denote the embedding model, where $\phi(x) \in \mathbb{R}^d$ maps a data point $x$ into a $d$-dimensional feature space. The model $\phi$ is trained on a labeled dataset $\mathcal{D} = \{(x_i, y_i)\}_{i=1}^{|\mathcal{D}|}$, where $x_i$ denotes a data point and $y_i$ its associated label. The learning of $\phi$ follows empirical risk minimization (ERM):

$$\min_{\phi, \theta} \ \mathbb{E}_{(x,y) \sim \mathcal{D}}[L(\theta(\phi(x)), y)],$$

where $\theta$ is a trainable classifier mapping the embedding $\phi(x)$ to label $y$, and $L$ is the loss function. Using the learned embedding model $\phi$, data points in the support set $(x_{s,i} \in \mathcal{S})$ and query set $(x_{q,j} \in \mathcal{Q})$ are transformed into their feature representations $\phi(x_{s,i})$ and $\phi(x_{q,j})$.

**Optimal Transportation.** Optimal transportation (OT) aims to realign distributions by minimizing the cost of transporting one distribution to another. This technique addresses discrepancies between datasets, enhances model generalization from training to testing, and yields more robust feature representations [7, 21]. A key concept in OT is the transport cost, which quantifies the effort required to move probability mass between distributions, often measured using metrics such as the Wasserstein distance.

Suppose there are finite samples in both the support set $x_{s,i} \in \mathcal{S}$ and the query set $x_{q,j} \in \mathcal{Q}$. Discrete OT employs empirical distributions to approximate the probability measures,

$$\hat{\mu}_s = \sum_i \delta_{s,i}, \quad \hat{\mu}_q = \sum_j \delta_{q,j}, \tag{1}$$

where $\delta_{s,i}$ and $\delta_{q,j}$ denote Dirac distributions. The discrete OT problem can then be formulated as

$$\pi^* = \arg\min_\pi \sum_{x_{s,i} \sim \hat{\mu}_s, \ x_{q,j} \sim \hat{\mu}_q} w(x_{s,i}, x_{q,j}) \, \pi(x_{s,i}, x_{q,j}), \tag{2}$$

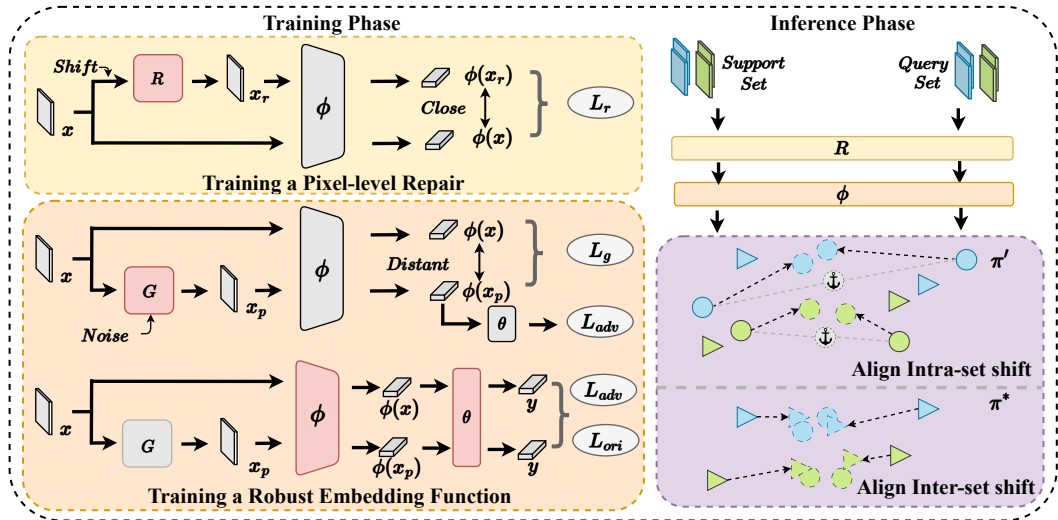

Figure 2: Overview of DUAL. In the training phase, we train a pixel-level repairer and a robust embedding function, which are then utilized during inference. In the inference phase, the objective is to align both intra-set and inter-set shifts using $\pi'$ and $\pi^*$.

where $w(x_{s,i}, x_{q,j})$ is the ground cost between samples.

To compute the transport plan $\pi^*$, Sinkhorn's algorithm [8] is often applied. With the optimal plan, the embeddings of the query set $\phi(x_{q,j})$ are transported to $\hat{\phi}(x_{q,j})$ via barycentric mapping [7], adapting the query set to the support set:

$$\hat{\phi}(x_{q,j}) = \frac{\sum_{x_{s,i} \in \mathcal{S}} \pi^*(x_{s,i}, x_{q,j})\, \phi(x_{s,i})}{\sum_{x_{s,i} \in \mathcal{S}} \pi^*(x_{s,i}, x_{q,j})}, \tag{3}$$

where $\hat{\phi}(x_{q,j})$ denotes the transported embedding of $x_{q,j}$. This allows the distance metric $M(\phi(x_{s,i}), \hat{\phi}(x_{q,j}))$ to be accurately computed within a shared embedding space.

**Dual Support-Query Shift in FSL.** Conventional FSL typically assumes a domain shift between the training and testing phases, i.e., $\mathcal{D}_{\text{Train}} \cap \mathcal{D}_{\text{Test}} = \varnothing$. However, in the meta-testing phase, an additional challenge arises: the support set $\mathcal{S}$ and the query set $\mathcal{Q}$ within each task may themselves be drawn from different distributions, denoted as $\mathcal{D}_S$ and $\mathcal{D}_Q$. As a result, the embeddings of support samples $\phi(x_s)$ and query samples $\phi(x_q)$ may lie in different spaces, leading to an *inter-set shift* that causes misclassification [3, 21]. This phenomenon, known as the *Support-Query Shift (SQS)* problem [21, 3, 1], has been widely studied. Yet, prior work primarily focuses on inter-set differences while overlooking variations within each set. We refer to this more general and challenging scenario as the *Dual Support-Query Shift (DSQS)* problem. In addition to inter-set shifts, DSQS accounts for intra-set shifts, where different instances within the same set (e.g., $q_i, q_j \in \mathcal{Q}$) may originate from distinct distributions. A similar issue arises in the support set. Moreover, under DSQS, the distributions of instances across support and query sets can also be mutually disjoint due to multiple unknown shifts, making alignment particularly difficult.

## 4 DUal ALignment Framework (DUAL)

To address DSQS, we present the **DUal ALignment Framework (DUAL)** for both training and inference. The key idea of DUAL is that clean features facilitate more reliable OT alignment. We first motivate DUAL by showing that both inter-set and intra-set shifts may mislead the OT plan (§4.1). To mitigate this issue, DUAL first obtains clean features through a pixel-level repairer and an adversarially trained embedding function (§4.2). It then reduces shifts using a dual-regularized OT framework, aligning instances to handle both inter-set and intra-set variations (§4.3).

In the training phase (*Left Part* of Fig. 2), we develop a pixel-level repairer $R$ and a robust embedding function $\phi$ for inference. Trainable network components are highlighted in pink. The yellow block illustrates corrupting the original data $x$ with shifts, which are then repaired by $R$. The repaired data $x_r$ is compared with $x$ using cosine similarity $L_r$ to optimize $R$. In the orange block, $L_g$ denotes negative cosine similarity, while $L_{adv}$ and $L_{ori}$ (Eq. (11)) represent KL divergence losses. A generator $G$ perturbs $x$ to produce a hard example $x_p$, which is less similar in the embedding space but remains in the same class. These hard examples are generated via $L_{adv}$ and $L_g$, while $L_{adv}$ and $L_{ori}$ jointly train the embedding function $\phi$.

In the inference phase (*Right Part* of Fig. 2), features of support and query samples are extracted by $R$ and $\phi$. As in Fig. 1, colors denote classes, while circles and triangles represent support and query features. Transported support features (dashed circles) are aligned with class-wise centroids (gray dashed circles with anchor symbols) according to the OT plan $\pi'$. Subsequently, transported query features (green dashed circles) are obtained through the OT plan $\pi^*$.

## 4.1 Motivation

While earlier OT-based FSL methods [21, 3] assume a *single* domain on each side (one-to-one alignment), the DSQS setting involves multiple domains in both the support and query sets. Following [30], we model each domain as a component of a Gaussian mixture, with the overall distribution represented as a class-weighted sum of Gaussians.

**Assumption 1** (DSQS Gaussian-mixture)**.** *For every class $c \in \mathcal{C}$, the latent feature $\phi(x) \in \mathbb{R}^d$ follows $\phi(x) \sim \mathcal{N}(\mu_{c,\diamond}, \Sigma_{c,\diamond})$, where $\diamond \in \{s, q\}$ denotes the* support *or* query *domain. The class prior $P(c)$ is shared across domains, but the component means $\mu_{c,\diamond}$ and covariances $\Sigma_{c,\diamond}$ may differ.*

Aggregating over classes yields the global moments

$$\mu_\diamond = \sum_c P(c)\,\mu_{c,\diamond}, \quad \Sigma_\diamond = \sum_c P(c)\big(\Sigma_{c,\diamond} + (\mu_{c,\diamond} - \mu_\diamond)(\mu_{c,\diamond} - \mu_\diamond)^\top\big). \tag{4}$$

Let $W_2(\mathcal{S}, \mathcal{Q})$ denote the 2-Wasserstein distance between the empirical feature distributions of the support set $\mathcal{S}$ and the query set $\mathcal{Q}$.

**Proposition 1** (OT cost under first-order Gaussian approximation)**.** *Approximating each domain by its first-order moments gives*

$$W_2^2(\mathcal{S}, \mathcal{Q}) \;=\; \underbrace{\|\mu_s - \mu_q\|_2^2}_{\text{inter-set mean gap}} \;+\; \underbrace{\text{tr}\big(\Sigma_s + \Sigma_q - 2(\Sigma_s^{1/2}\Sigma_q\Sigma_s^{1/2})^{1/2}\big)}_{\text{inter-set covariance gap}}. \tag{5}$$

Thus, the transport cost grows monotonically with (i) the inter-set mean gap $\|\mu_s - \mu_q\|_2$, and (ii) the intra-set spreads $\text{tr}(\Sigma_s)$ and $\text{tr}(\Sigma_q)$.[1]

**Proposition 2** (Error of transported embeddings)**.** *Let $\hat{\phi}(x_{q,i})$ be the transported query embedding obtained from the clean OT plan in Eq. (3), and $\hat{\phi}_\sigma(x_{q,i})$ its noisy counterpart. Assume additive Gaussian noise $\eta \sim \mathcal{N}(0, \sigma_\diamond^2 I)$ is independently injected in both support and query domains $\diamond \in \{s, q\}$. Then,*

$$\mathbb{E}\big[\|\hat{\phi}(x_{q,i}) - \hat{\phi}_\sigma(x_{q,i})\|_2^2\big] = d\big(\sigma_s^2 + \sigma_q^2\big).$$

Higher noise levels $\sigma_s, \sigma_q$ therefore *increase* the risk of mismatched OT plans, ultimately degrading classification accuracy.

Summing up, Propositions 1 and 2 highlight two key sources of error under DSQS: **(i) domain misalignment** (mean/covariance gaps), and **(ii) feature noise**. Our DUAL framework addresses both: (i) it contracts domain gaps via dual-regularized OT, and (ii) it learns a noise-tolerant embedding, thereby reducing $W_2(\mathcal{S}, \mathcal{Q})$ and stabilizing transported features.

---

[1] If $\Sigma_s$ and $\Sigma_q$ commute, a common high-dimensional approximation [6], the cross term vanishes, reducing the trace expression to $\text{tr}(\Sigma_s + \Sigma_q)$.

## 4.2 Dual Adversarial Training

During training, we adopt a two-level adversarial strategy to obtain clean features for subsequent dual alignment, thereby enhancing model robustness by suppressing input noise and improving embedding quality. Specifically, we first train a repairer $R$ to remove noise from input images. Then, we adversarially train an embedding function $\phi$ using a generator $G$, which produces "hard" examples to strengthen its robustness.

**Training a Pixel-level Repairer.** To mitigate pixel-level noise, we first train a repairer $R$ to restore shifted images, thereby helping $\phi$ extract cleaner features. The key idea of training $R$ is to minimize the embedding-space distance of $\phi$ between features before and after repair. In this way, $R$ preserves the original semantic structure while reducing noise. Notably, $R$ provides $\phi$ with low-noise inputs, which theoretically tightens the feature-noise upper bound $\sigma' \leq L\varepsilon_p$ (detailed in Lemma 1).

As illustrated in the yellow block of Fig. 2, we add a predefined shift to the original data $x$, and the repairer network $R$ generates the restored data $x_r$. The training objective is:

$$\min_{\phi} \mathbb{E}_{x \sim \mathcal{D}} \Big[ \min_{x_r} M(\phi(x_r), \phi(x)) \Big], \tag{6}$$

where $M$ denotes the comparison metric in FSL, such as Euclidean distance or cosine similarity. We encourage $\phi(x_r)$ to be *closer* to $\phi(x)$ so that $R$ learns to correct the imposed shift, producing cleaner representations that reduce semantic distortion during inference.

To train $R$, we minimize the embedding-space distance between the repaired data $x_r$ and the original data $x$, formulated as

$$\min_{R} M(\phi(x_r), \phi(x)). \tag{7}$$

In this way, the repairer $R$ learns to correct diverse shifts with a single model, showing that our framework is a bias-agnostic solution applicable to real-world scenarios.

**Training a Robust Embedding Function.** After corrupted data are restored by the repairer $R$, we further enhance the robustness of the embedding function $\phi$ through adversarial training with *hard examples* generated by a network $G$. As shown in the orange block (upper part) of Fig. 2, $G$ is trained to produce perturbed samples $x_p$ that are *less similar* to the original data point $x$ in the embedding space, by maximizing the comparison loss. To make $\phi$ resilient to such perturbations, we adopt the following minmax objective:

$$\min_{\phi} \mathbb{E}_{x \sim \mathcal{D}} \Big[ \max_{x_p} M(\phi(x_p), \phi(x)) \Big]. \tag{8}$$

In practice, we sample a batch of augmented candidates $\{x_p\}$ and select the one that maximizes the loss $L$, so that $\phi$ is updated against the hardest instance. As illustrated in the orange block (bottom part) of Fig. 2, we realize $G$ as a semantic-aware generator:

$$x_p = G(x) \text{ s.t. } \|\theta(\phi(x_p)) - \theta(\phi(x))\|_2^2 \leq \epsilon, \tag{9}$$

where $G$ perturbs $x$ into $x_p$ while preserving its class semantics. We adopt dropout [14] for stochasticity. Unlike conventional adversarial training that perturbs inputs via i.i.d. noise (e.g., $x_p \sim \mathcal{N}(x, \sigma^2 I)$) [39, 47], our generator encodes semantic structure directly, requiring fewer samples to achieve convergence [13].

We enlarge the embedding distance between $x$ and its perturbed counterpart $x_p$. To ensure that $G$ retains sufficient class semantics, we enforce that the generated example $x_p$ can still be classified as the same label $y$, using KL divergence [37] as a regularizer. In practice, we adopt the soft-label vector form of $y$ for the KL term. The generator is therefore trained with the following objective:

$$\max_{G} \ M(\phi(G(x)), \phi(x)) \ - \ KL\big(\theta(\phi(G(x))), y\big). \tag{10}$$

Following [21], we optimize both $G$ and $R$ via stochastic gradient descent (SGD). During this stage, the parameters of the embedding functions $\phi$ and $\theta$ are kept fixed [2].

Once the hard examples are derived, we train the embedding functions to acquire more robust features. As shown in the orange block (bottom part) of Fig. 2, we jointly minimize the empirical risk of both the original data $x$ and the perturbed example $x_p$ using KL divergence:

$$\min_{\phi,\theta} \; \lambda \underbrace{KL(\theta(\phi(x)), y)}_{L_{ori}} + (1 - \lambda) \underbrace{KL(\theta(\phi(x_p)), y)}_{L_{adv}}, \tag{11}$$

where $\lambda$ controls the trade-off. During inference, DUAL extracts robust features through the repairer $R$ and the embedding function $\phi$. While their combination may appear straightforward, our design enforces a clear separation of roles: $R$ acts as a pixel-level denoiser, whereas $\phi$ operates at the feature level. Since $G$ challenges $\phi$ by generating semantically perturbed samples, $R$ must instead preserve proximity to the original semantic space. Jointly training $R$ and $\phi$ would thus lead to conflicting objectives and hinder convergence, making the decoupled design crucial for stability.

### 4.3 Dual Regularized Optimal Transportation

Previous OT-based methods for SQS assume a single-domain alignment, which becomes insufficient under the DSQS setting, as highlighted in Proposition 1. To address this, we propose a *dual regularized optimal transportation* scheme for inference. After obtaining clean features from the repairer $R$ and robust embedding $\phi$, we extend classical OT with negative-entropy regularization to stabilize the transport plan.[2]

**Intra-set Alignment via Regularized OT.** As illustrated in the purple block of Fig. 2, we first perform intra-set alignment by transporting support samples $\mathcal{S}$ to their class-wise centroids $\overline{\mathcal{S}}$ (gray dashed circles with anchor symbols), yielding a plan $\pi'$ and a transported support set $\mathcal{S}' = \{x'_{s,i}\}$. This step reduces intra-class variance and alleviates intra-set shifts. Formally,

$$\pi' = \arg\min_{\pi} \sum_{\substack{x_{s,i} \in \mathcal{S} \\ \overline{x}_{s,k} \in \overline{\mathcal{S}}}} \beta \, w(x_{s,i}, \overline{x}_{s,k}) \pi(x_{s,i}, \overline{x}_{s,k}) + (1 - \beta) \, \pi(x_{s,i}, \overline{x}_{s,k}) \log \pi(x_{s,i}, \overline{x}_{s,k}), \tag{12}$$

where $\overline{\mathcal{S}}$ denotes class-wise centroids and $\beta$ controls the smoothness of the transport plan.

**Inter-set Alignment with Anchored OT.** After obtaining $\pi'$ and the transported support set $\mathcal{S}'$, we align the query set $\mathcal{Q}$ with $\mathcal{S}'$. Specifically, we build the queryanchor cost matrix

$$C_{j,i}^{\mathcal{Q},\mathcal{S}'} = w\big(\phi(x_{q,j}), \, \phi(x'_{s,i})\big), \tag{13}$$

where $w(\cdot, \cdot)$ is the ground cost used in Eq. (12). We then reuse the same regularized OT formulation, replacing $(\mathcal{S}, \overline{\mathcal{S}})$ with $(\mathcal{Q}, \mathcal{S}')$, and obtain $\pi^*$ via Sinkhorn scaling.

In summary, the dual OT scheme produces two transport plans, $\pi'$ (supportcentroid) and $\pi^*$ (query-support), which jointly mitigate intra-set and inter-set shifts under DSQS. Notably, the first-stage alignment is only relevant for multi-shot cases; in the one-shot setting, no intra-class centroid alignment is required.

## 5   Theoretical Analysis

Here, we analyze four key quantities that characterize the behavior of the DUAL framework: (i) the post-repair noise variances $\sigma'_s$ and $\sigma'_q$; (ii) the class-conditional covariances $\Sigma_{c,\diamond}$ for $\diamond \in \{s, q\}$; (iii) the 2-Wasserstein distance $W_2(\mathcal{S}, \mathcal{Q})$ between the aligned support and query distributions; and (iv) the classification risk $\Pr[h(x) \neq y]$ under a 1-Lipschitz nearest-prototype classifier $h$. Detailed proofs can be found in Appendix A.1.

We first show that variance contraction is achieved by the repairer.

**Lemma 1** (Variance contraction). *Assume that: 1) $\phi$ is L-Lipschitz, i.e., $\|\phi(u) - \phi(v)\|_2 \leq L\|u - v\|_2$; 2) The repair network $R$ satisfies an expected pixel-space MSE of $\varepsilon_p^2 = \mathbb{E}_x \|R(x) - x\|_2^2$. Then the post-repair feature noise in each domain satisfies $\sigma'_\diamond \leq L \varepsilon_p$, where $\diamond \in \{s, q\}$.*

---

[2] In the one-shot scenario, where only one sample exists in the support set, we apply regularized optimal transport to align the support set and query set.

Lemma 1 shows that pixel-space denoising reduces feature-space noise linearly via the Lipschitz continuity of $\phi$, thereby controlling stochastic variation within domains. To prevent adversarial shifts from inflating class-conditional spreads, we next establish a margin enlargement result.

**Lemma 2** (Margin enlargement). *Assume dual adversarial training is used with a target margin parameter $\kappa > 0$, following a margin-based objective such as:*

$$\min_G \max_\phi \Big[ -\cos\big(\phi(x), \phi(G(x))\big) + \lambda\, \mathcal{L}_{\text{cls}}\big(G(x), y\big) \Big], \tag{14}$$

*where $\mathcal{L}_{\text{cls}}$ (e.g., CE or KL divergence) penalizes label changes. Then for any $x$, $\|\phi(x) - \phi(G(x))\|_2 \geq \kappa$, and $G(x)$ preserves the class label of $x$. Consequently, $\text{tr}\big(\Sigma_{c,\diamond}\big) - \text{tr}\big(\Sigma_{c,\diamond}^{\text{adv}}\big) \geq \kappa^2$, for all $c$ and $\diamond \in \{\text{s}, \text{q}\}$.*

While Lemma 1 controls random noise, Lemma 2 guarantees a deterministic margin between clean and adversarial features of the same class, thereby contracting each class-conditional covariance ellipsoid. Together, the two lemmas imply that the combined effect of Repair $R$ and Generator $G$ tightens the geometry of both domains. We now quantify how this geometric tightening reduces the 2-Wasserstein distance between the support and query distributions.

**Theorem 1** (Contracted OT bound). *Under the assumptions of Lemmas 1–2, define $\kappa_T = \kappa$, $\varepsilon_T = \varepsilon_p$, $\rho_T = e^{-\beta t}$ for $t$ Sinkhorn iterations with damping $\beta > 0$. Then,*

$$\mathbb{E}\big[W_2^2(\mathcal{S}, \mathcal{Q})\big] \leq (1 - \rho_T)\Big[\underbrace{\|\mu_{\text{s}} - \mu_{\text{q}}\|_2^2 - 2\kappa_T}_{\text{shrunk mean gap}} + \underbrace{\text{tr}(\Sigma_{\text{s}} + \Sigma_{\text{q}}) - 2\kappa_T^2}_{\text{shrunk covariance}}\Big] + 2L\sqrt{d}\,\varepsilon_T. \tag{15}$$

Eq. (15) reveals that the support-query transport cost contracts by (at least) $2\kappa_T$ in the means and $2\kappa_T^2$ in the covariances, up to a vanishing solver residual $\rho_T$ and the small repair term $L\varepsilon_T$. A reduced Wasserstein distance, in turn, strengthens the generalization guarantees of Lipschitz classifiers. The next corollary makes this connection explicit.

**Corollary 1** (Classification risk). *Let $h$ be a 1-Lipschitz nearest-prototype classifier in the aligned space, and define $\Delta = \min_{c \neq c'} \|\mu_c - \mu_{c'}\|_2$. If $\Delta > 2\kappa_T$, then under the same assumptions as Theorem 1, the classification risk satisfies*

$$\Pr[h(x) \neq y] \leq \frac{W_2^2(\mathcal{S}, \mathcal{Q})}{\Delta^2} + \exp\Big(-\frac{\kappa_T^2}{2(L\varepsilon_T)^2}\Big). \tag{16}$$

Eq. (16) decomposes the error into a *distribution mismatch term*, $W_2^2(\mathcal{S}, \mathcal{Q})/\Delta^2$, and a *robustness term* that decays exponentially with the squared margin $\kappa_T^2$. Hence, the alignment strategies simultaneously minimise domain divergence and enlarge the safety margin around each class prototype, yielding provably lower risk.

Summing up, Lemma 1 establishes that pixel-level repair reduces feature noise via Lipschitz continuity, while Lemma 2 shows that dual adversarial training enforces feature separation and contracts class-conditional covariances. Building on these, Theorem 1 proves that DUAL reduces the Wasserstein distance between support and query distributions after repair and alignment, up to a solver residual. Finally, Corollary 1 bounds the classification error, revealing that generalization is jointly governed by inter-class separation and the robustness margin. Together, these results provide a rigorous foundation for how DUAL achieves robust alignment and lowers classification risk under domain shift.

# 6 Experiment

We evaluate DUAL against 10 state-of-the-art baselines on three public datasets. Due to space limitations, the pseudo-codes, dataset details, baseline descriptions, and implementation details are provided in Appendix B.

**Setup.** To validate our framework, we evaluate on three standard benchmark datasets: **(1)** CIFAR-100 [23], **(2)** mini-ImageNet [44], and **(3)** Tiered-ImageNet [38] for FSL. We compare against 10 state-of-the-art FSL methods, divided into three categories: **(i)** four conventional FSL baselines: MatchingNet [45], ProtoNet [42], TransPropNet [31], and FTNet [9]; **(ii)** three support-query

Table 1: Quantitive results of DUAL.

| Methods | CIFAR-100 | mini-ImageNet | Tiered-ImageNet | CIFAR-100 | mini-ImageNet | Tiered-ImageNet |
|---|---|---|---|---|---|---|
| | 1-shot | | | 5-shot | | |
| MatchingNet [45] | $30.26_{\pm0.38}$ | $43.62_{\pm0.47}$ | $30.01_{\pm0.41}$ | $40.35_{\pm0.33}$ | $56.24_{\pm0.37}$ | $35.05_{\pm0.36}$ |
| ProtoNet [42] | $28.53_{\pm0.30}$ | $43.84_{\pm0.44}$ | $30.15_{\pm0.41}$ | $41.59_{\pm0.41}$ | $59.83_{\pm0.42}$ | $43.41_{\pm0.43}$ |
| TransPropNet [31] | $31.01_{\pm0.34}$ | $24.22_{\pm0.29}$ | $24.18_{\pm0.32}$ | $37.06_{\pm0.40}$ | $25.93_{\pm0.29}$ | $35.48_{\pm0.37}$ |
| FTNet [9] | $22.36_{\pm0.21}$ | $37.04_{\pm0.44}$ | $22.01_{\pm0.30}$ | $26.19_{\pm0.25}$ | $49.14_{\pm0.36}$ | $24.50_{\pm0.23}$ |
| AQ [12] | $35.86_{\pm0.54}$ | $31.59_{\pm0.44}$ | $30.24_{\pm0.40}$ | $53.93_{\pm0.49}$ | $43.85_{\pm0.49}$ | $38.54_{\pm0.42}$ |
| TP [3] | $30.89_{\pm0.42}$ | $45.66_{\pm0.55}$ | $29.34_{\pm0.43}$ | $45.50_{\pm0.37}$ | $62.32_{\pm0.38}$ | $41.92_{\pm0.39}$ |
| PGADA [21] | $34.90_{\pm0.45}$ | $50.37_{\pm0.57}$ | $28.47_{\pm0.40}$ | $49.45_{\pm0.38}$ | $61.09_{\pm0.39}$ | $40.73_{\pm0.34}$ |
| AQP [1] | $31.68_{\pm0.39}$ | $30.59_{\pm0.43}$ | $30.40_{\pm0.40}$ | $45.09_{\pm0.46}$ | $42.65_{\pm0.57}$ | $45.34_{\pm0.60}$ |
| RAS [35] | $36.98_{\pm0.38}$ | $50.40_{\pm0.32}$ | $31.05_{\pm0.40}$ | $50.02_{\pm0.21}$ | $63.95_{\pm0.40}$ | $43.98_{\pm0.42}$ |
| SSL-ProtoNet [28] | $36.00_{\pm0.38}$ | $28.59_{\pm0.30}$ | $29.31_{\pm0.48}$ | $48.74_{\pm0.37}$ | $36.56_{\pm0.32}$ | $35.65_{\pm0.37}$ |
| DUAL-P | $38.93_{\pm0.50}$ | $53.00_{\pm0.60}$ | $34.29_{\pm0.50}$ | $54.47_{\pm0.40}$ | $67.83_{\pm0.40}$ | $47.81_{\pm0.41}$ |
| DUAL-M | $39.35_{\pm0.51}$ | $54.44_{\pm0.59}$ | $35.29_{\pm0.37}$ | $50.11_{\pm0.40}$ | $64.04_{\pm0.42}$ | $42.96_{\pm0.38}$ |

Table 2: Ablation studies and In-depth analysis of DUAL.

| Techniques Variants | CIFAR-100 | mini-ImageNet | Tiered-ImageNet | CIFAR-100 | mini-ImageNet | Tiered-ImageNet |
|---|---|---|---|---|---|---|
| | 1-shot | | | 5-shot | | |
| w./o. dual AT & OT | $27.43_{\pm0.32}$ | $43.93_{\pm0.47}$ | $27.85_{\pm0.35}$ | $41.97_{\pm0.41}$ | $63.60_{\pm0.45}$ | $40.48_{\pm0.40}$ |
| w./o. dual AT | $31.36_{\pm0.41}$ | $53.43_{\pm0.59}$ | $30.76_{\pm0.43}$ | $42.00_{\pm0.44}$ | $66.22_{\pm0.46}$ | $40.84_{\pm0.40}$ |
| w./o. dual OT | $34.63_{\pm0.40}$ | $40.88_{\pm0.45}$ | $27.54_{\pm0.36}$ | $53.20_{\pm0.44}$ | $66.69_{\pm0.43}$ | $30.57_{\pm0.34}$ |
| w./o. $G$ | $35.98_{\pm0.28}$ | $43.74_{\pm0.79}$ | $29.32_{\pm0.37}$ | $47.10_{\pm0.47}$ | $61.22_{\pm0.78}$ | $43.95_{\pm0.49}$ |
| w./o. $R$ | $27.47_{\pm0.36}$ | $44.12_{\pm0.43}$ | $26.73_{\pm0.26}$ | $35.05_{\pm0.39}$ | $62.33_{\pm0.38}$ | $37.92_{\pm0.32}$ |
| $Fixed\ G$ | $38.48_{\pm0.50}$ | $55.35_{\pm0.61}$ | $31.12_{\pm0.47}$ | $52.47_{\pm0.47}$ | $66.91_{\pm0.47}$ | $42.54_{\pm0.40}$ |
| Enc shift to $\phi$ | $34.56_{\pm0.38}$ | $49.37_{\pm0.50}$ | $24.26_{\pm0.26}$ | $45.98_{\pm0.38}$ | $62.55_{\pm0.39}$ | $29.11_{\pm0.29}$ |
| TP + $R$ | $32.03_{\pm0.36}$ | $48.58_{\pm0.53}$ | $28.52_{\pm0.39}$ | $46.13_{\pm0.40}$ | $64.25_{\pm0.40}$ | $41.22_{\pm0.38}$ |
| DUAL-P | $38.93_{\pm0.50}$ | $53.00_{\pm0.59}$ | $34.29_{\pm0.50}$ | $54.47_{\pm0.40}$ | $67.83_{\pm0.40}$ | $47.81_{\pm0.41}$ |

shift baselines: TP [3], PGADA [21], and AQP [1]; **(iii)** three adversarially robust FSL baselines: RAS [35], AQ [12], and SSL-ProtoNet [28]. Since DUAL is a model-agnostic adversarial alignment framework, we implement it with different classifiers, e.g., ProtoNet (DUAL-P) and MatchingNet (DUAL-M).

**Quantitative results.** Table 1 shows that DUAL consistently outperforms the four conventional baselines (MatchingNet, ProtoNet, TransPropNet, and FTNet), achieving an average improvement of 24.16%. These methods fail to address the distribution shift between support and query sets, which DUAL effectively realigns using adversarial training. Compared to adversarially robust FSL methods, DUAL (including DUAL-P and DUAL-M) relatively surpasses SSL-ProtoNet by 35.65% on average by aligning distributions at both the task and instance levels. Although TP and AQP also leverage optimal transport, they remain sensitive to small perturbations and relatively suffer 12.93% and 21.86% accuracy loss, respectively. Overall, DUAL achieves up to 25.66% higher accuracy than state-of-the-art methods on average by reconstructing information lost due to instance-level shifts.

**Ablation Studies.** We conduct ablation studies on DUAL-P with ProtoNet (similar trends hold for MatchingNet). As shown in Table 2, removing both dual adversarial training (Dual AT) and dual regularized optimal transport (Dual OT) causes a substantial performance drop, confirming their necessity. Specifically, Dual AT improves accuracy by an average of 11.59%, with the largest gain observed on Tiered-ImageNet in the 1-shot setting (from 30.76% to 34.29%), where clean features are crucial for reliable alignment (see §4.1). Adding Dual OT further enhances performance, with improvements of up to 17.24% in 5-shot accuracy on Tiered-ImageNet, as it explicitly mitigates distribution shifts in the feature space through optimal transport.

**In-depth Analysis.** We further analyze the roles of the generator ($G$) and repairer ($R$). Removing $G$ relatively reduces accuracy by 11.80%, showing its importance in generating adversarial perturbations for robustness. Excluding $R$ causes a 22.54% relative drop, highlighting its critical role in repairing features for alignment. Fixing $G$ during training relatively lowers performance by 3.20%, indicating that a trainable $G$ better captures instance-specific variations. Training only the encoder $\phi$ on perturbed images without $R$ yields a 7.06% gain, but still underperforms the full model. Adding $R$ on TP [3] can boost performance by 5.24%, confirming its role in mitigating shifts and improving generalization. These results validate the complementary roles of $G$ and $R$ in robust few-shot learning under domain shifts. Due to space limitations, we provide additional visualization effects in the appendix, comparing cosine similarity to illustrate how features contribute to alignment. For

Table 3: The results of adopting DSQS and CD-FSL.

| Method | In-Domain | Out-of-Domain | | | |
|---|---|---|---|---|---|
| | ImageNet-1K | Aircraft | Describable Textures | Fungi | MSCOCO |
| ProtoNet [42] | $21.14_{\pm 0.55}$ | $20.12_{\pm 0.47}$ | $35.51_{\pm 0.49}$ | $18.82_{\pm 0.60}$ | $21.41_{\pm 0.67}$ |
| TransPropNet [31] | $10.91_{\pm 0.25}$ | $11.86_{\pm 0.29}$ | $19.85_{\pm 0.28}$ | $10.13_{\pm 0.31}$ | $10.15_{\pm 0.31}$ |
| AQ [12] | $15.94_{\pm 0.50}$ | $19.53_{\pm 0.40}$ | $30.11_{\pm 0.29}$ | $18.09_{\pm 0.61}$ | $20.46_{\pm 0.20}$ |
| PGADA [21] | $22.67_{\pm 0.50}$ | $17.91_{\pm 0.48}$ | $36.04_{\pm 0.47}$ | $22.72_{\pm 0.63}$ | $26.69_{\pm 0.64}$ |
| DUAL-P | $25.66_{\pm 0.54}$ | $21.83_{\pm 0.57}$ | $40.69_{\pm 0.51}$ | $26.28_{\pm 0.69}$ | $31.66_{\pm 0.71}$ |
| Method | | Out-of-Domain | | | |
| | Omniglot | Quick Draw | VGG Flower | CUB-200-2011 | Traffic Signs |
| ProtoNet [42] | $17.09_{\pm 0.56}$ | $28.43_{\pm 0.73}$ | $47.98_{\pm 0.72}$ | $24.94_{\pm 0.63}$ | $29.32_{\pm 0.74}$ |
| TransPropNet [31] | $10.15_{\pm 0.29}$ | $10.63_{\pm 0.31}$ | $14.70_{\pm 0.42}$ | $11.85_{\pm 0.35}$ | $10.90_{\pm 0.31}$ |
| AQ [12] | $12.98_{\pm 0.11}$ | $18.21_{\pm 0.36}$ | $49.60_{\pm 0.72}$ | $26.33_{\pm 0.31}$ | $20.38_{\pm 0.68}$ |
| PGADA [21] | $32.81_{\pm 0.82}$ | $40.97_{\pm 0.73}$ | $49.93_{\pm 0.73}$ | $24.71_{\pm 0.60}$ | $31.63_{\pm 0.68}$ |
| DUAL-P | $50.71_{\pm 0.92}$ | $53.58_{\pm 0.77}$ | $60.45_{\pm 0.76}$ | $31.17_{\pm 0.73}$ | $37.33_{\pm 0.74}$ |

example, on mini-ImageNet, adopting intra-OT and inter-OT increases the alignment similarity by approximately 14.6% and 20.5%, respectively.

**When DSQS Meets Cross-Domain FSL.** Cross-Domain (CD) FSL introduces a domain gap between the training and testing sets, whereas DSQS imposes dual shifts at meta-test time: inter-set shifts between support and query sets, and intra-set shifts within each set. To evaluate both settings in a unified framework, we conduct experiments on four representative baselines using Meta-Dataset, which spans ten public image datasets across diverse domains. Following the protocol of [44], we meta-train on the ImageNet-1K training split and evaluate on ImageNet-1K (in-domain) as well as the remaining datasets (out-of-domain). Numbers of ways, shots, and query images are randomly sampled as in [16]. As shown in Table 3, DUAL-P achieves the best performance across all domains. These results indicate that DUAL generalizes effectively under both in-domain and out-of-domain conditions, and remains robust to the dual shifts characteristic of DSQS.

# 7 Discussion and Conclusion

**Real-world Complex Tasks.** DUAL can be applied to complex real-world tasks such as quality monitoring and beverage deterioration monitoring [19, 20], where distribution shifts are common. By aligning distributions in the embedding space, our framework is able to maintain robustness in these settings. For example, adapting DUAL to object detection or segmentation requires only modifications to the training objective, such as the choice of loss functions. The key concept of DUAL is first to extract clean features via Dual AT and then perform improved alignment through Dual OT, which together address both inter-set and intra-set shifts. Pre-trained models such as CLIP can also provide cleaner features due to their strong generalization capabilities [25], but they introduce higher computational costs and slower inference in real-world applications.

**Limitations and Future Work.** Overall, this work introduced DSQS as a challenging FSL scenario characterized by both inter-set and intra-set distribution shifts, and proposed the *DUal ALignment Framework (DUAL)* to mitigate them. Theoretical and empirical results demonstrate that DUAL outperforms ten baselines across multiple datasets. Looking ahead, we envision extending DUAL to broader vision tasks, exploring stronger embedding functions, and investigating additional techniques for addressing DSQS. Nevertheless, DUAL still leaves room for refinement and integration of more advanced techniques. Incorporating recent advances in robust embedding learning or distribution alignment could further improve its effectiveness. For example, more sophisticated adversarial training strategies [4] or advanced OT formulations [40, 32] may better capture the nuances of real-world data distributions.

# Acknowledgment

This work was supported by the National Science and Technology Council (NSTC), Taiwan, the Ministry of Education (MOE), Taiwan, under Grants NSTC 114-2223-E-002-009, NSTC 114-2221-E-002-180-MY3, MOE 114L895504, MOE 114L9009, Research Grants Council (RGC) of Hong Kong, China, under the General Research Fund (GRF) Grant No. 14203420 and Collaborative Research Fund (CRF) Grant No. C1045-23G.

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

# A Proof

## A.1 Preliminary: Optimal Transportation

**Episode data and embeddings**  Let the support set be $S = \{x_{s,i}\}_{i=1}^m$ with labels $\{y_{s,i}\}$ and the query set be $Q = \{x_{q,j}\}_{j=1}^n$. The embedding model $\phi : \mathcal{X} \to \mathbb{R}^d$ yields features $z_{s,i} = \phi(x_{s,i})$ and $z_{q,j} = \phi(x_{q,j})$. Unless otherwise stated, we use uniform masses $a_i = \frac{1}{m}$ and $b_j = \frac{1}{n}$ for support and query, collected as $a \in \Delta_m$ and $b \in \Delta_n$.

**Ground cost and cost matrix**  Let $w(\cdot, \cdot)$ be a ground cost between two features (e.g., squared Euclidean $w(u, v) = \|u - v\|_2^2$ or cosine distance $w(u, v) = 1 - \frac{\langle u, v \rangle}{\|u\|_2 \|v\|_2}$). The episode cost matrix is $C \in \mathbb{R}^{m \times n}$ with entries
$$C_{i,j} = w(z_{s,i}, z_{q,j}).$$

**Discrete Kantorovich OT (primal form)**  A transport plan is a nonnegative matrix $\pi \in \mathbb{R}_+^{m \times n}$ that moves probability mass from $S$ to $Q$ while satisfying marginal constraints:
$$\pi \mathbf{1}_n = a, \qquad \pi^\top \mathbf{1}_m = b.$$
The (unregularized) OT problem minimizes the total cost
$$\min_{\pi \geq 0} \langle C, \pi \rangle \quad \text{s.t. } \pi \mathbf{1}_n = a, \ \pi^\top \mathbf{1}_m = b,$$
which upper-bounds the squared 2-Wasserstein distance between the empirical feature distributions of $S$ and $Q$.

**Entropy-regularized OT and Sinkhorn scaling**  For stability with few samples and noisy features, we adopt an entropy-regularized objective consistent with our framework:
$$\min_{\pi \geq 0} \beta \langle C, \pi \rangle + (1 - \beta) \sum_{i,j} \pi_{i,j} \log \pi_{i,j} \quad \text{s.t. } \pi \mathbf{1}_n = a, \ \pi^\top \mathbf{1}_m = b,$$
where $\beta \in (0, 1]$ trades off fidelity to $C$ (large $\beta$) and smoothness of $\pi$ (small $\beta$). This is equivalent to the common form $\langle C, \pi \rangle + \tau \sum_{i,j} \pi_{i,j} (\log \pi_{i,j} - 1)$ with temperature $\tau = \frac{1 - \beta}{\beta}$. Defining the Gibbs kernel $K = \exp(-C/\tau)$ elementwise, the optimizer is obtained by Sinkhorn iterations:
$$v \leftarrow b \oslash (K^\top u), \quad u \leftarrow a \oslash (Kv), \quad \pi = \operatorname{diag}(u) \, K \, \operatorname{diag}(v),$$
where $\oslash$ denotes elementwise division.

**Barycentric projection (feature transport)**  Given an optimal plan $\pi$, we align query features to the support geometry via barycentric mapping:
$$\hat{z}_{q,j} = \frac{\sum_{i=1}^m \pi_{i,j} \, z_{s,i}}{\sum_{i=1}^m \pi_{i,j}},$$
and compute the metric $M(\cdot, \cdot)$ (e.g., cosine or Euclidean) in the aligned space for classification.

In our two-stage alignment (§4.3), we instantiate this machinery twice: (i) an intra-set plan $\pi'$ that transports support instances to class-wise centroids, producing $S'$, and (ii) an inter-set plan $\pi^*$ that aligns queries to $S'$.

## A.2 Proof of Proposition 1

*Proof.* When both of the marginals $\mu_s, \mu_q$ are Gaussian distributions, the problem can be greatly simplified. A closed-form solution exists. Denote the mean and covariance of $\mu_*$ and $\Sigma_*$, respectively. Let $S, Q$ be two Gaussian random vectors associated with $\mu_s, \mu_q$, respectively. Then, the cost becomes
$$\mathbb{E}\{\|S - Q\|^2\} = \mathbb{E}\{\|\tilde{S} - \tilde{Q}\|^2\} + \|m_s - m_q\|^2, \tag{17}$$
where $\tilde{S} = S - m_s$ and $\tilde{Q} = Q - m_q$ are the zero-mean versions of $S$ and $Q$. We minimize (17) over all possible Gaussian joint distributions between $X$ and $Y$, resulting in

$$\min_{K} \left\{ \|m_s - m_q\|^2 + trace(\Sigma_s + \Sigma_q - 2K) \ \Big| \ \begin{bmatrix} \Sigma_0 & K \\ K^T & \Sigma_1 \end{bmatrix} \geq 0 \right\},$$

with $K = \mathbb{E}\{\tilde{S}\tilde{Q}^T\}$. The constraint is semidefinite, so the above problem is a semidefinite programming (SDP). It turns out that the unique minimizer in closed form achieves the minimum

$$K = \Sigma_s^{1/2}(\Sigma_s^{1/2}\Sigma_q\Sigma_s^{1/2})^{1/2}\Sigma_s^{-1/2}$$

with minimum value

$$W(\mu_s, \mu_q)^2 = \|m_0 - m_1\|^2 + trace(\Sigma_s \tag{18}$$

$$+\Sigma_q - 2(\Sigma_s^{1/2}\Sigma_q\Sigma_s^{1/2})^{1/2}). \tag{19}$$

The consequent displacement interpolation $\mu_t$ is a Gaussian distribution with mean $m_t = (1 - t)m_s + tm_q$ and covariance

$$\Sigma_t = \Sigma_s^{-1/2}\left((1-t)\Sigma_q + t(\Sigma_s^{1/2}\Sigma_q\Sigma_s^{1/2})^{1/2}\right)^2 \Sigma_s^{-1/2}. \tag{20}$$

$\square$

## A.3 Proof of Proposition 2

We first provide a Lemma to prove the Proposition 2.

**Lemma 3.** *The error of the transportation cost is*

$$W_\sigma(\mu_s, \mu_q) \leq W(\mu_s, \mu_q) \leq W_\sigma(\mu_s, \mu_q) + \sqrt{d(\sigma_s^2 + \sigma_q^2)},$$

*where* $W_\sigma(\mu_s, \mu_q) := W(\mu_s * \mathcal{N}_{\sigma_s}, \mu_q * \mathcal{N}_{\sigma_q})$ *denotes the original support and query set distribution $\mu_s$ and $\mu_q$ being perturbed with Gaussian noises $\sigma_s$ and $\sigma_q$.*

Note that $|\cdot|$ is the absolute value, and $*$ is the convolution operator. Based on Lemma 3, we estimate the error of transported embedding $\hat{\phi}(x_{s,i})$ in Eq. (3).

*Proof.* The left-hand side inequality immediately follows because $W$ is non-increasing under convolutions, since $\mathcal{N}_{\sqrt{\sigma_s^2 + \sigma_q^2}} = \mathcal{N}_{\sigma_s} * \mathcal{N}_{\sigma_q}$, where $*$ is the convolution operator.

On the right side of the inequality, we adopt Kantorovich-Rubinstein duality to write the optimal transport as follows.

$$W(\mu_s, \mu_q) = \sup_{\|w\|_{Lip} \leq 1} E_{\mu_s}[w] - E_{\mu_q}[w] \tag{21}$$

$$W_\sigma(\mu_s, \mu_q) = \sup_{\|w\|_{Lip} \leq 1} E_{\mu_s * \mathcal{N}_{\sigma_s}}[w_\sigma] - E_{\mu_q * \mathcal{N}_{\sigma_q}}[w_\sigma] \tag{22}$$

where $\|\cdot\|_{Lip}$ is the Lipschitz norm. Letting $w^*$ be optimal for $W(\mu_s, \mu_q)$, we obtain,

$$W_\sigma(\mu_s, \mu_q) = E_{\mu_s * \mathcal{N}_{\sigma_s}}[w^*] - E_{\mu_q * \mathcal{N}_{\sigma_q}}[w^*]. \tag{23}$$

Let $X_s \sim \mu_s$, $Z_s \sim N_{\sigma_s}$ as independent random variables, we have,

$$|E_{\mu_s}[w^*] - E_{\mu_s * N_{\sigma_s}}[w^*]| \tag{24}$$
$$= E[w^*(X_s)] - E[w^*(X + Z_s)]$$
$$\leq E[\|Z_s\|_2^2] = \sqrt{d}\sigma_s.$$

where the last inequality uses $\|w^*\|_{Lip} \leq 1$. $d$ is the dimension of the embedding vector. Similarly, $X_q \sim \mu_q$, $Z_q \sim N_{\sigma_q}$ as independent random variables, we have,

$$|E_{\mu_q}[w^*] - E_{\mu_q * N_{\sigma_q}}[w^*]| \tag{25}$$
$$= E[w^*(X_q)] - E[w^*(X + Z_q)]$$
$$\leq E[\|Z_q\|_2^2] = \sqrt{d}\sigma_q.$$

By inserting Eq. (24) and Eq. (25) to Eq. (23), and Cauchy-Schwarz inequality, we concludes the proof. $\square$

In the following, we prove the proposition.

*Proof.* Base on Lemma 3, barycentric coordinate is defined as follows,

$$\hat{\pi}_i^* = \frac{\pi^*(x_{s,i}, x_{q,j})}{\sum_{x_{q,j} \in \mathcal{Q}} \pi^*(x_{s,i}, x_{q,j})} \sim \mathcal{N}_{\sigma_s} \tag{26}$$

Let $X_q \sim \mu_q$, $X_q^\sigma \sim \mu_q * N_{\sigma_q}$ as independent random variables,

$$E[X_q^{\sigma(t)} - X_q^{(t)}] = \sigma_q, \tag{27}$$

where $X_q^{\sigma(t)}$ and $X_q^{(t)}$ denotes the $t$-th dimension of random variable $X_q^\sigma$ and $X_q$, respectively.

Combining Eq. (26) and Eq. (27), the perturbed distribution $\hat{X}_s \sim \mu_s * N_{\sigma_s} * N_{\sigma_q} = \mu_s * N_{\sqrt{\sigma_s^2 + \sigma_q^2}}$.

$$E[\hat{X}_q - X_q] = \sqrt{d(\sigma_s^2 + \sigma_q^2)}. \tag{28}$$

As the noise level, i.e., $\sigma_s$, and $\sigma_q$, increases, it is more likely to mislead the transportation plan and alleviate the model's performance.

$\square$

## A.4    Proof of Lemma 1

*Proof.* Fix $\diamond = $ s (the query case is identical). The noisy feature of a sample is $\tilde{z} = \phi(R(x))$, and the noisefree feature is $z = \phi(x)$. By Lipschitz continuity, $\|\tilde{z} - z\|_2 \leq L \|R(x) - x\|_2$. Squaring and taking the expectation over $x \sim \mathcal{D}_s$ yields:

$$\mathbb{E}\|\tilde{z} - z\|_2^2 \leq L^2 \, \mathbb{E}\|R(x) - x\|_2^2 = L^2 \, \varepsilon_p^2.$$

The left-hand side is $\sigma_s'^2$, so $\sigma_s' \leq L\varepsilon_p$.

$\square$

## A.5    Proof of Lemma 2

*Proof.* The optimality of the generator $G$ follows from the fact that, for a fixed encoder $\phi$, the cosine loss remains strictly larger than its minimum when $\|\phi(x) - \phi(G(x))\|_2 < \kappa$; thus, the margin constraint must be met to avoid increasing the classification loss $\mathcal{L}_{cls}$. Conversely, given an optimal generator, the encoder $\phi$ seeks to maximize the cosine loss by pushing $\phi(G(x))$ away from $\phi(x)$, thereby enforcing the margin $\kappa$. This adversarial separation has the effect of tightening the classconditional covariance: letting $\mu_{c,\diamond} = \mathbb{E}[\phi(x) \mid y = c]$ denote the class center, adding adversarial examples $\phi(G(x))$ at distance $\kappa$ decreases the empirical second moment around $\mu_{c,\diamond}$ by at least $\kappa^2$.

$\square$

## A.6    Proof of Theorem 1

*Proof.* We derive the bound by considering the combined effects of mean alignment, covariance reduction, feature noise, and Sinkhorn solver inaccuracy. First, dual-adversarial training enforces a minimum margin $\kappa_T$ between each sample and its adversarial counterpart in feature space. As a result, each centroid (i.e., domain mean) moves toward the other by up to $\kappa_T$, leading to at least a $2\kappa_T$ reduction in the squared Euclidean distance between the support and query means due to the reverse triangle inequality. Similarly, since adversarial samples are at least $\kappa_T$ away from their respective class centers, the empirical second moment around each class center is reduced by at least $\kappa_T^2$, and across both domains, the total covariance trace is decreased by at least $2\kappa_T^2$. Next, the feature noise introduced by the pixel-level repair network is bounded in expectation by $L\varepsilon_T$ per dimension, and across a $d$-dimensional embedding, this contributes an additional distortion bounded by $2L\sqrt{d}\,\varepsilon_T$ when considering one sample from each domain. Finally, the entropic Sinkhorn solver used to approximate the optimal transport cost yields a $(1 - \rho_T)$-contracted estimate of the true cost, where $\rho_T = e^{-\beta t}$ depends on the number of iterations $t$ and regularization strength $\beta$ [11]. Combining these effects yields the desired bound in Eq. (15).

$\square$

## A.7 Proof of Corollary 1

*Proof.* We derive the classification risk bound by considering two failure modes: transport error and repair noise. Let $x$ be a query sample and $x'$ its aligned prototype. Since $h$ is 1Lipschitz, the classification decision satisfies

$$|h(x) - h(x')| \leq \|\phi(x) - \phi(x')\|_2, \tag{29}$$

and by Markovs inequality, the probability that $h(x) \neq h(x')$ is bounded by $\mathbb{E}[\|\phi(x) - \phi(x')\|_2^2]/\Delta^2$, where $\Delta$ is the minimum inter-class prototype separation.

This yields the first term. For the second term, Lemma 1 implies that the repaired feature $\phi(R(x))$ deviates from the true feature $\phi(x)$ by a sub-Gaussian variable with parameter $L\varepsilon_T$, so the probability that this deviation exceeds the margin $\kappa_T$ is bounded by $\exp(-\kappa_T^2/2(L\varepsilon_T)^2)$ via Hoeffdings inequality. A misclassification occurs if either the transport error moves the sample outside its class region or the repair noise shifts the feature beyond the margin; by the union bound, this yields the combined upper bound in Eq. (16). □

# B Implementation Details

## B.1 Pseudo Code of DUAL.

---
**Algorithm 1** DUal Alignment Framework (DUAL)

---
**Require:** Training dataset $\mathcal{D}$, comparison module $M$, learning rate $\eta$, trade-off parameters $\lambda_1$ and $\lambda_2$, an arbitrary shift $S$, support set $\mathcal{S}$, query set $\mathcal{Q}$.
**Ensure:** Embedding model $\phi$, repairer $R$, Optimal Plan $\pi^*$,
1: Initialize generator $G$, repairer $R$,
2: Initialize embedding model $\phi$, classifier $\theta$.
3: *# Training Stage*
4: **for** $\{x, y\}$ in $\mathcal{D}$ **do**
5:    *# fixed $\phi, \theta$, update G, R*
6:    $x_p = G(x), x_c = S(x), x_r = R(x_c)$
7:    $L_g = -M(\phi(x_p), \phi(x)), L_r = M(\phi(x_r), \phi(x)),$
8:    $L_{adv} = KL(\theta(\phi(x_p)), y))$
9:    *# Generated less similar data points.*
10:    $G \leftarrow G - \eta\nabla(L_g + L_{adv}), \ R \leftarrow R - \eta\nabla L_r$
11:    *# fixed G, R, update $\phi, \theta$*
12:    $x_p = G(x),$
13:    $L_{ori} = KL(\theta(\phi(x)), y)), \ L_{adv} = KL(\theta(\phi(x_p)), y))$
14:    *# classifying the generated samples correctly.*
15:    $\phi \leftarrow \phi - \eta\nabla(\lambda L_{ori} + (1-\lambda)L_{adv})$
16:    $\theta \leftarrow \theta - \eta\nabla(\lambda L_{ori} + (1-\lambda)L_{adv})$
17: *# Inference Stage*
18: Solve the Eq. 12 to obtain $\pi'$ and $\pi^*$
19: $\mathcal{S}_f = \phi(R(\mathcal{S})), \mathcal{Q}_f = \phi(R(\mathcal{Q}))$
20: $\overline{\mathcal{S}}_f = \frac{1}{|\mathcal{S}_f|}\sum \mathcal{S}_f, \mathcal{S}'_f = \pi'(\mathcal{S}_f, \overline{\mathcal{S}}_f), \mathcal{Q}'_f = \pi^*(\mathcal{S}'_f, \mathcal{Q}_f)$
21: **for** $\{f'_s, f'_q, y_s\}$ in $\mathcal{S}'_f, \mathcal{Q}'_f$ **do**
22:    $y_q = M(f_s, f_q, y_s)$

---

## B.2 Details of Datasets

- **CIFAR-100** consists of $60,000$ three-channel square images of size $32 \times 32$, evenly distributed in 100 classes. Classes are evenly distributed in 20 superclasses. We employ 19 image transformations [51], each being applied with 5 different intensity levels, to evaluate the robustness of a model.

- **mini-ImageNet** contains $60,000$ square images with three channels of size $224 \times 224$ from the ImageNet dataset with a 64-classes training set, a 16-classes validation set, and a 20-classes test set [45]. Similar to CIFAR-100, mini-ImageNet also has the same transformations proposed by [15] to simulate different domains [15].

- **Tiered-ImageNet** [38] contains 779,165 three-channel $84 \times 84$ images, grouped into 34 higher-level nodes in 608 classes. The nodes are partitioned into 20, 6, and 8 disjoint sets of training, validation, and testing nodes, and the corresponding classes form the respective meta-sets.

- **Meta-dataset** [44] contains 10 public image datasets of a diverse range of domains: ImageNet-1k, Omniglot, FGVCAircraft, CUB-200-2011, Describable Textures, QuickDraw, FGVCx Fungi, VGG Flower, Traffic Signs, and MSCOCO. Each dataset has train/val/test splits.

### B.3   Details of Baselines

- **MatchingNet [45]** measures the pairwise cosine similarity between the support set and the query set and assigns the same class of the support example to the query example.

- **ProtoNet [42]** uses Euclidean distance to classify queries to the prototype embeddings, i.e., averaging the embeddings of all support examples in the same class.

- **TransPropNet [31]** is an extension of ProtoNet, which utilizes a graph neural network of labels, leveraging information about local neighborhoods.

- **FTNET [9]** is a meta-learning framework that estimates the distribution between the training and testing sets transductively.

- **AQ [12]** is a robust FSL baseline designed to produce adversarially robust meta-learners and investigate the causes of adversarial vulnerability.

- **TP [3]** combines the ProtoNet, optimal transport, and transductive batch normalization to solve the support-query shift in few-shot learning.

- **PGADA [21]** reduces optimal transportation errors by learning from self-supervised hard examples and using negative entropy regularization.

- **AQP [1]** aims to create more challenging virtual query sets by adversarially perturbing the query sets, inducing a distribution shift between support and query sets. AQP can be regarded as the SOTA method in support-query shift few-shot learning using episodic training.

- **SSL-ProtoNet [28]** is a metric-based few-shot learning approach that combines self-supervised learning, Prototypical Networks, and knowledge distillation to leverage sample discrimination effectively.

- **RAS [35].** is a robust FSL baseline that employs high-level feature matching between base class data without the need for adversarial samples.

### B.4   Details of Implementation

Following [3], we report the average top-1 accuracy score with a $95\%$ confidence interval from 2000 runs. In addition, we conduct the tasks of 1-shot and 5-shot with 16-target, i.e., 1 or 5 instances per class in the support set and 16 instances in the query set, in CIFAR-100, mini-ImageNet, and Tiered-ImageNet. Same with [21], we use a 4-layer convolutional network as an embedding function $\phi$ on CIFAR-100, ResNet18 for mini-ImageNet, and Tiered-ImageNet. As a general adversarial training framework for few-shot learning, we combine DUAL-P with two classifiers, i.e., ProtoNet [42] and MatchingNet [45], in the testing phase. As for our repairer $R$, we adopted an adjusted REDNET-like[34] structure, composed of a 4-layer encoder-decoder structure and 2 convolutional layers. In practice, we also adopt optimal transport [21] and self-supervised learning by deploying the NT-Xent Loss [5] on unlabeled data from the testing set. The learning rate $\eta$, batch size $b$, and embedding dimention $d$ are set to $1e-3$, 128, 128, respectively. Besides, SGD with Adam optimizer [22] is adopted to train the model in 200 epochs with early stopping. Grid search is adopted to select the trade-off parameter in the objective function, i.e., $\lambda = 0.5$, $\beta = 0.5$ for best performance. Note that, for simplicity, most AQP baselines are evaluated under the SQS setting for comparison since DSQS is a harder setting for AQP. In addition, we re-implemented RAS for evaluation. Also, we adopt the transductive batch normalization [31] on TransPropNet, FTNET, TP, PGADA, and our framework. Most of experiments are conducted on a workstation equipped with a NVIDIA GeForce RTX 4090 GPU (24 GB), an Intel Core i9-14900K CPU, and 128 GB of memory.

Table 4: Cosine similarity of visualizations in different datasets. Higher is better.

| Dataset | Original | Add Shifts | Adopting intra-OT | Adopting inter-OT |
|---------|----------|-----------|-------------------|-------------------|
| CIFAR-100 | 0.6954 | 0.5076 | 0.6338 | 0.7224 |
| mini-ImageNet | 0.6519 | 0.6083 | 0.6971 | 0.7332 |
| Tiered-ImageNet | 0.6758 | 0.6117 | 0.6515 | 0.7038 |

## C  Discussions

### C.1  Visualization Impact of Multiple Shifts

We provide a visualization of the impact on multiple shifts. As shown in Fig. 3, we can see that real-world images frequently display multiple shifts within the same set. In particular, the multiple shifts, such as blur, noise, weather, and digital distortions, impact image quality and classification performance. Individually, each shift degrades the image, but combined shifts (e.g., Blur + Noise or All) significantly distort the visual features, making the image more challenging to interpret. These compounded shifts enlarge the data distribution and pose greater challenges for models, especially when the shifts are unpredictable or unknown, leading to reduced robustness and accuracy. This highlights the importance of addressing multiple shifts to improve model performance.

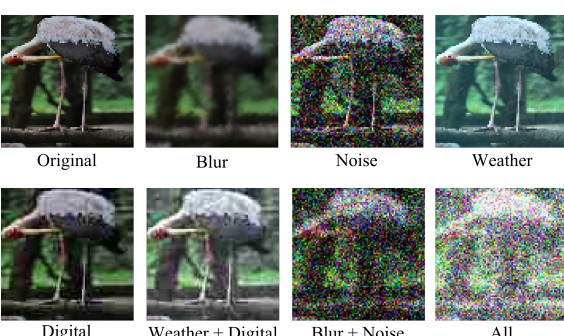

Figure 3: Visualization of the impact on multiple shifts. Multiple shifts make the image harder to classify and enlarge the distribution compared to the original image, especially when such shifts are unknown.

### C.2  Visualization Analysis in Embedding Space

To quantify how our method mitigates inter- and intra-distribution shifts, we report cosine distance, i.e., cosine similarity as a alternative visualization between support and query embeddings before and after applying the method, rather than relying on large-scale t-SNE plots, which are unstable and sensitive to hyperparameters.. Specifically, we select large-scale samples from the CIFAR-100, mini-ImageNet, and Tiered-ImageNet datasets to construct support and query sets from different classes and extract their embeddings to compare the cosine similarity of these samples. The key difference in this comparison is whether or not dual optimal transportation (intra-OT and inter-OT) is applied. As shown in Table 4, we observe that the distances are closer when dual OT is used, demonstrating that our approach effectively reduces the distribution shift. We believe these experiments provide comprehensive evidence supporting our conclusions. In particular, applying intra-OT recovers a meaningful portion of the lost consistency but does not fully return to the original level. Inter-OT provides the most robust recovery, outperforming intra-OT on every dataset and narrowing the gap to the original the most. The effect is especially pronounced on the more heterogeneous dataset, suggesting that aligning relationships across samples is particularly effective when variability is higher.

### C.3  Computation Overhead of DUAL

We conduct experiments on the average time of computational overhead for each component in one epoch of 5-way 1-shot. As shown in Table 5, the computational overhead analysis highlights the significant variation in training and testing time across datasets and methods. For training, CIFAR-10 is the least computationally demanding, requiring only 0.52 hours for 100 epochs, compared to 18.50 and 14.72 hours for mini-ImageNet and Tiered-ImageNet, respectively, indicating the higher complexity of the latter datasets. During testing, the per-epoch time remains minimal for all datasets, with CIFAR-10 being the fastest at 0.003 hours, followed by 0.009 hours at Tiered-ImageNet and

0.045 hours at mini-ImageNet. Notably, the computational cost is justified by the accuracy improvements observed in the corresponding methods, suggesting that the overhead remains acceptable for practical applications. Future work should focus on optimizing these methods to further reduce the time complexity without compromising performance.

Table 5: Computation Cost of DUAL in Training and Inference Phase . Time in the training phase denotes the wall-clock time of 100 epochs (Hours). Time in the inference phase denotes the inference time in each epoch (Hours).

| Methods | CIFAR-100 | mini-ImageNet | Tiered-ImageNet |
|---------|-----------|---------------|-----------------|
| Training Cost | | | |
| $R$ | 23% | 29% | 27% |
| $G$ | 15% | 33% | 32% |
| $\phi$ | 62% | 38% | 41% |
| Time | 0.52 | 18.50 | 14.72 |
| Inference Cost | | | |
| Dual OT | 14% | 3% | 11% |
| $R$ | 28% | 8% | 11% |
| Time | 0.003 | 0.045 | 0.009 |

## C.4 Broader Impact

DUAL tackles the critical challenge of Dual Support Query Shift (DSQS) in few-shot learning (FSL), significantly enhancing the alignment of distributions under both inter-set and intra-set shifts. By delivering robust performance in highly dynamic and unpredictable environments, DUAL has the potential to make machine learning systems more adaptable, resource-efficient, and accessible. These advancements hold promising applications in fields such as healthcare and autonomous driving. However, the adversarial training employed in DUAL, while designed for robustness, could potentially inspire misuse in crafting adversarial attacks on other machine learning models. Researchers and practitioners should remain vigilant about ensuring ethical use of such techniques. Overall, DUAL represents a step forward in making few-shot learning more robust and capable of handling real-world challenges. Its broader impact lies in improving the reliability, adaptability, and accessibility of AI systems across diverse domains. However, it is essential to remain mindful of the ethical and environmental considerations associated with the framework, encouraging responsible research and deployment practices.

