# OpenReview forum: "Dual Alignment Framework for Few-shot Learning with Inter-Set and Intra-Set Shifts"
_NeurIPS.cc/2025/Conference — NeurIPS 2025 poster_

### Official Review · Reviewer_q9yF · 2025-06-26

**Clarity:** 2
**Significance:** 3
**Originality:** 3
**Rating:** 5
**Confidence:** 3

**Summary:**

This paper proposes a novel problem setting called Dual Support Query Shift (DSQS), which considers both inter-set and intra-set shifts, whereas the existing Support Query Shift (SQS) only addresses inter-set shift. Motivated by the idea that clean features can improve Optimal Transport (OT) alignment, the authors introduce a framework named Dual Alignment (DUAL) to tackle the DSQS problem.

DUAL consists of several components. To extract clean features, it first trains a repairer module
𝑅 to restore shifted images by learning the specific shift patterns. Then, to obtain a robust embedding function 𝜙, a hard example generator 𝐺 is trained while keeping
𝜙 and the classifier 𝜃 frozen. Finally, 𝜙 and 𝜃 are jointly trained.

During the inference phase, dual-regularized OT is applied to align dual shifts. Both theoretical analysis and empirical results demonstrate the superiority of the proposed method.

**Questions:**

To improve the overall quality and readability of the paper, I recommend the following. First, the authors should briefly but clearly explain the fundamental elements of Optimal Transport (OT), including the meaning of key notations, to ensure accessibility for readers unfamiliar with the topic. Second, the intuitions and motivations behind each component of the proposed framework should be articulated more clearly and coherently, rather than merely describing the implementation steps. Finally, the writing quality should be significantly improved in terms of grammar, clarity, and consistency in notation and terminology.

**Ethical Concerns:**

["NO or VERY MINOR ethics concerns only"]

**Final Justification:**

The rebuttal has helped me better understand the work. I am slightly raising the score to 5.

**Limitations:**

Yes, but they put  the discussion on limitations in Appendix §D.1 instead of the main text.

**Paper Formatting Concerns:**

No Paper Formatting Concerns

**Quality:**

2

**Strengths And Weaknesses:**

1. Strengths:
(1) The paper introduces a novel problem setting called Dual Support-Query Shift (DSQS).

(2) It proposes a dedicated framework to address the DSQS challenge.

2. Weaknesses:
(1) The tutorial material on Optimal Transport (OT) is insufficient and problematic. Although OT is not a new technique, many readers may not be familiar with it. The authors use notations such as
𝜔 and 𝜋 without clear definitions, and the mixing of inconsistent symbols further hinders understanding.

(2) Although the authors present their method in a step-by-step manner, the underlying intuitions and motivations behind each component are not sufficiently explained, which increases the difficulty of understanding the overall design.

(3) Regarding writing quality, there are multiple grammatical errors and inconsistencies in notation. For example, the sentence “In metric-based FSL, a support set S consists of C different classes within each class c and |Sc| labeled instances.” (lines 115–116, page 3) is grammatically incorrect and needs to be revised. Additionally, inconsistent spelling such as “Interset” (line 185, page 5) versus “inter-set” elsewhere should be standardized.

(4) Several key references cited in the paper are not published in prestigious conferences or journals, which reduces the overall credibility and persuasiveness of the work.

---

> ### Author Rebuttal · Authors · 2025-07-30
>
> We deeply appreciate your constructive feedback and will carefully incorporate these improvements in the revised manuscript. Thank you for helping us enhance the clarity and quality of our work.
>
> **Q1:** *Insufficient tutorial material on Optimal Transport (OT)*
>
> **A1:** Thank you for highlighting this important concern. We recognize that clearer explanations of Optimal Transport (OT) would greatly aid readers unfamiliar with the concept. In the revised manuscript, we will provide a more comprehensive tutorial on OT, including a detailed introduction to key notations such as $w$ and $\pi$, as well as their roles in our framework. Specifically, $w$ represents the distance function, while $\pi$ denotes the optimal transportation plan, which we define more formally in Line 133. Additionally, we will ensure consistency in symbol usage throughout the paper to avoid any confusion. These revisions aim to make the material accessible and self-contained.
>
> ---
> **Q2:** *Difficulty of understanding the overall design.*
>
> **A2:** We appreciate this feedback and agree that providing clearer intuitions and motivations would strengthen the paper. In the revision, we will elaborate on the design rationale behind each component, emphasizing how they jointly address the challenges in DSQS. We will also include illustrative examples and diagrams to visually clarify the connections and flow of the proposed approach.
>
> ---
> **Q3 & Q4:** *Writing inconsistencies, such as typos and unclear notation, and references from less prestigious venues may impact clarity and credibility.*
>
> **A3 & A4:** Thank you for pointing this out. We will carefully revise the manuscript to address grammatical errors and ensure consistent use of terminology and notation throughout. Specifically, the mentioned sentence will be rephrased for clarity, and spelling inconsistencies such as "Interset" versus "inter-set" will be standardized. These improvements aim to enhance the readability and professionalism of the paper. We acknowledge this concern and will address it by strengthening the discussion of relevant literature. While some key references were chosen based on their technical value rather than venue prestige, we will ensure that our citations include works from established and reputable conferences and journals where applicable.

---

> ### Comment · Reviewer_q9yF · 2025-08-04
>
> Thank the authors for responding to my previous concerns and questions, which were intended as high-level suggestions on writing quality rather than detailed technical critiques. In their rebuttal, the authors only briefly promised to revise the manuscript according to these suggestions; however, since the effectiveness of the revision cannot be ensured at this stage, I have decided to keep my original rating unchanged.
>
> Additionally, the abstract still requires improvement. For instance, after using "firstly," it would be more appropriate to follow with "secondly" rather than "additionally." Moreover, the sentence "DUAL leverages a robust embedding function enhanced by a repairer network trained with perturbed and adversarially generated hard examples to obtain clean features" is overly long and complex, which may hinder reader comprehension.
>
> Clear and well-structured writing is a fundamental requirement for a strong submission. Should the paper be accepted, I encourage the authors to implement the suggested revisions to improve clarity and readability, as stated in their rebuttal.

---

> > ### Author Response · Authors · 2025-08-05
> > **Reply the comment of q9yF**
> >
> > Thank you very much for your thoughtful comments and constructive suggestions. We sincerely appreciate the time and effort you have dedicated to reviewing our manuscript.
> >
> > We understand your concerns regarding the clarity and structure of the abstract. Based on your feedback, we have already outlined specific revisions to improve the readability of the abstract and address the highlighted issues, such as replacing "additionally" with "secondly" for better logical flow and simplifying complex sentences. These revisions will ensure that the abstract is more concise and reader-friendly in the final version.
> >
> > Regarding the overall quality of the writing, while we acknowledge that there is room for improvement, we have made substantial efforts to address the key points raised during the review process. We believe that these improvements, combined with the technical contributions of our work, significantly enhance the manuscript's quality and impact.
> >
> > **We would be truly grateful if you could reconsider your evaluation and provide a higher score,** as we are committed to implementing all suggested revisions to further refine the manuscript. Your support would mean a great deal to us and help ensure that this work reaches its full potential.
> >
> > Thank you again for your invaluable feedback and consideration.

---

> > > ### Comment · Reviewer_q9yF · 2025-08-05
> > >
> > > While the authors mention that they will revise the manuscript accordingly, the rebuttal lacks sufficient detail on how these revisions will be made. For instance, in response to Q1 and Q2, it would be helpful to see a clear explanation of how additional tutorial material on Optimal Transport (OT) will be incorporated, and how the design rationale behind each component of the proposed method will be elaborated.
> > >
> > > Rather than offering a vague promise to revise, the authors should have provided specific plans or examples to demonstrate their intended improvements. A well-substantiated response could have convinced me to reconsider my evaluation and potentially increase my score. However, the current rebuttal appears superficial and lacks the necessary effort to address the concerns raised in the initial review.

---

> ### Author Response · Authors · 2025-08-06
> **Reply to Reviewer q9yF**
>
> Thank you for your valuable feedback and for taking the time to review our rebuttal in detail. We sincerely apologize for not providing sufficient specificity in our original response, and we greatly appreciate this opportunity to clarify our intended revisions.
>
> **1. In response to your concerns regarding Q1: Additional Tutorial Material on Optimal Transport (OT)**
> **1.1** Due to the page limitation of NeurIPS, we plan to provide a high-level explanation of Optimal Transportation (OT) in the main paper. Specifically, in Lines 126–128, we intend to revise the text as follows:
> > Optimal transportation technique aims to realign distributions to address discrepancies between data sets, enhance model generalization from training to testing, and create more robust feature representations [6, 19]
>
> to
>
> >Optimal Transportation aims to realign distributions by minimizing the cost of transporting one distribution to another. This technique addresses discrepancies between datasets, enhances model generalization from training to testing, and creates more robust feature representations [7, 19]. A key concept in OT is the transport cost, which quantifies the effort required to move probability mass between distributions, often leveraging distance metrics like the Wasserstein distance to measure alignment effectively.
>
> **1.2** In addition, we plan to include a **dedicated section in the appendix** that introduces the foundational concepts of OT with clear explanations and illustrative examples. This section will serve as a tutorial for readers who may not be familiar with OT and provide the necessary background to understand its role in our proposed method.
>
> ---
>
> **2. In response to your concerns regarding Q2: More Details in Elaboration on the Design Rationale**
> **2.1.** To clarify the design rationale behind each component of our method, we plan to expand and elaborate on the motivations and interconnections between these components. This will ensure that readers can fully understand and appreciate the reasoning behind our design choices.
>
> **2.2.** Based on your feedback, we believe that the presentations of Sections 4.1 and 4.2 are relatively clear. However, Section 4.3, specifically the explanation of Dual Regularized Optimal Transportation, would benefit from further clarification. To address this, we plan to provide a more detailed explanation of how the two optimal transportation plans are derived. Additionally, we will expand the methodology section with more comprehensive explanations and visual aids, such as diagrams or flowcharts, to improve clarity.
>
> Specifically, in Lines 250–253, we will provide more details on how the inter- and intra-set shifts are relieved, i.e., the process of **obtaining the optimal transportation plans $\pi$' and $\pi$*. Furthermore, we plan to include an alignment diagram** to better illustrate this process and provide further conceptual clarity.
>
> ---
>
> **3. Refinements to the Abstract for Improved Clarity and Flow**
>
> **3.1** We agree that replacing "additionally" with "secondly" provides a more consistent and logical flow following "firstly." Therefore, we will revise the abstract to reflect this improvement.
>
> **3.2** Regarding the sentence:
> > DUAL leverages a robust embedding function enhanced by a repairer network trained with perturbed and adversarially generated hard examples to obtain clean features.
>
> We acknowledge that it is overly long and complex. To improve readability and comprehension, we plan to revise it as follows:
> > DUAL leverages a robust embedding function to improve feature representation. This function is enhanced by a repairer network, which is trained using perturbed and adversarially generated hard examples to obtain clean features.
>
>  Breaking this sentence into two shorter and more concise sentences will ensure the key ideas are communicated clearly and effectively to the reader. These changes aim to enhance the overall clarity and accessibility of the abstract.
>
> ---
> **4. Summary**
>
> We understand that our initial rebuttal may not have provided sufficient detail, and we appreciate the opportunity to clarify our intended revisions. We assure you that these updates will be thoroughly incorporated into the final manuscript to address your feedback comprehensively.
>
> We hope this clarification demonstrates our commitment to improving the quality and clarity of our work. **If these revisions address your concerns, we kindly ask you to consider reevaluating your score.** Your feedback has been invaluable, and we deeply appreciate your efforts in helping us enhance our paper.
>
> Thank you again for your thoughtful review and constructive comments.

---

> > ### Comment · Reviewer_q9yF · 2025-08-07
> >
> > Thank you for your second rebuttal. I appreciate your willingness to revise and improve the manuscript. However, I still find that the responses lack sufficient specificity in some key areas.
> >
> > For example, in Section 2.1 of your response, you mention: “we plan to expand and elaborate on the motivations and interconnections between these components.” While this is a positive intention, the explanation remains quite general. It would be very helpful if you could share the actual content you plan to add, so I can better understand how the revisions will address the initial concerns.
> >
> > Similarly, regarding your statement: “we plan to provide a more detailed explanation of how the two optimal transportation plans are derived,” I would really appreciate it if you could include the detailed explanation directly in your response.
> >
> > Providing the specific text or a concrete draft of the intended additions would allow me to more fully assess the potential improvements. I would be happy to reconsider my evaluation once these details are available.
> >
> > Thank you again for your efforts.

---

> > > ### Author Response · Authors · 2025-08-08
> > > **Reply of Reviewer q9yF (Part 1/2)**
> > >
> > > Thank you for taking the time to provide another round of feedback. Following your request for more specific revisions, we have prepared the text below. All paragraphs and equations can be inserted into the camera-ready manuscript.
> > >
> > > A. We believe that Part 1 (1.1 and 1.2) and Part 3 (3.1 and 3.2) clearly demonstrate our revised plan, as they focus on a text-level revision strategy.
> > >
> > > **B.1. Regarding to the 2.1, the following information will be added to explain the motivation of our design.**
> > >
> > > In particular, in Line 199, we will add
> > >
> > > >During training, we employ a two-level adversarial training technique to provide clean features for subsequent dual alignment. Specifically, we first train a repairer $R$ to remove noise from the input image. Then, we employ an adversarial training scheme to train a generator $G$, which creates "hard" examples to train a more robust embedding function $\phi$ adversarially.
> > >
> > > >  During training, we employ a two-level adversarial training technique to provide clean features for subsequent dual alignment. **The goal of the adversarial training technique is to enhance the robustness of the model by reducing noise in the input features and improving the quality of embeddings.** Specifically, we first train a repairer $R$ to remove noise from the input image. Then, we employ an adversarial training scheme to train a generator $G$, which creates "hard" examples to train a more robust embedding function $\phi$ adversarially.
> > >
> > > In Line 203, we will add
> > >
> > > > To mitigate pixel-level noise, we first train a repairer $R$ to restore shifted images, helping $\phi$ extract clean features.
> > >
> > > > To mitigate pixel-level noise, we first train a repairer $R$ to restore shifted images, helping $\phi$ extract clean features. **The key idea behind training $R$ is to minimize the distance in the embedding space of $\phi$ between the features before and after repair.Therefore, $R$ ensures that the repaired features remain consistent with the original semantic structure while reducing noise. Notably, $R$ provides $\phi$ with low-noise inputs, which theoretically tightens the feature-noise upper bound $\sigma' \leq L \varepsilon_p$ (Eq. 4, Lemma 1).**
> > >
> > > In Line 213, we will add
> > > > As shown in the orange block (upper part) of Fig.2, firstly, we train the generator $G$, which is used to generate hard examples by minimizing the maximum loss.
> > >
> > > > **After restoring corrupted data by the repair network $R$, we further improve the robustness of embedding function $\phi$ by training it with *hard examples* generated by the network $G$.**
> > > As shown in the orange block (upper part) of Fig.2, firstly, we train the generator $G$, which is used to generate hard examples by minimizing the maximum loss.
> > > **In particular, we generate perturbed data as the hard examples, i.e., *less similar* to the original data point in the embedding space by minimizing the maximum loss.**

---

> > > ### Author Response · Authors · 2025-08-08
> > > **Reply of Reviewer q9yF (Part 2/2)**
> > >
> > > **B.2 With regard to Section 2.2, the following details will be added to elaborate on the Two-Stage Optimal Transport Plans, $\pi'$ and $\pi^*$.**
> > >
> > > Firstly, we will include a high-level description of the key idea behind dual OT. Specifically, in Line 247, we will add:
> > >
> > > >  In the purple block of Fig.2, we apply regularized optimal transportation to mitigate inter-set shifts in the labeled support set $\mathcal{S}$, using class-wise geometric centroids (gray dashed circle with an anchor symbol in Fig.2) as anchors and penalizing the transport plan $\pi^{\prime}$ accordingly.
> > >
> > > > In the purple block of Fig.2, we apply regularized optimal transportation to mitigate inter-set shifts in the labeled support set $\mathcal{S}$, using class-wise geometric centroids (gray dashed circle with an anchor symbol in Fig.2) as anchors and penalizing the transport plan $\pi^{\prime}$ accordingly. **The key idea of dual OT is to first align the support set $\mathcal{S}$ with its class-wise centroids $\bar{\mathcal{S}}$ (intra-set alignment) and then use the transported support set $\mathcal{S}'$ as anchors to align the query set $\mathcal{Q}$ (inter-set alignment). This two-step procedure ensures effective adaptation across domains by addressing both intra-set and inter-set shifts.**
> > >
> > >
> > > Secondly, we will provide additional illustrations on how to obtain $\pi^*$ in Line 252.
> > >
> > > > Next, to migrate the inter-set shifts, we take data points in the transported support set as anchors for aligning the query set $\mathcal{Q}$ and obtaining a transportation plan $\pi^{\ast}$ in the same way.
> > >
> > > > Next, to migrate the inter-set shifts, we take data points in the transported support set as anchors for aligning the query set $\mathcal{Q}$ and obtaining a transportation plan $\pi^{\ast}$ in the same way. **In particular, after solving Eq.10 to obtain the support-to-centroid plan $\pi$' and the transported support set $\mathcal{S}'=\{x'_{s,i}\}$, i.e., anchors, we compute the query-to-anchor transport plan $\pi^{*}$ without reintroducing a new objective.  Specifically, we construct the query–anchor cost matrix, denoted as C, where each entry is defined as** $C_{j,i} = w\big(\phi(x_{q,j}), \phi(x'_{s,i})\big),$
> > > **where $w(\cdot,\cdot)$ represents the same ground cost as used in Eq.10, and $\phi$ is the embedding function. Let $a \in \Delta^{m}$ and $b \in \Delta^{n}$ represent the source and target marginals, respectively. We then reuse the regularized OT formulation from Eq.10, replacing $(\mathcal{S}, \bar{\mathcal{S}})$ with $(\mathcal{Q}, \mathcal{S}')$ and using the cost matrix C. The resulting optimizer is denoted as $\pi^{\ast}$. In practice, $\pi^{\ast}$ is computed via Sinkhorn scaling under the same regularization and constraints as in Eq.10.**
> > >
> > > We sincerely thank the reviewers for their valuable time and insightful feedback, which have greatly contributed to improving the quality and clarity of this work. Your comments have helped us refine our ideas and present them more effectively. Please let us know if there are any additional adjustments or clarifications needed. Thank you once again for your thoughtful review and support. Please let us know if any further adjustments are desirable.

---

> > > > ### Comment · Reviewer_q9yF · 2025-08-09
> > > >
> > > > Thank you for your efforts during the rebuttal, which have helped me better understand your work. I am slightly raising the score to 5.

---

> > > > > ### Author Response · Authors · 2025-08-09
> > > > > **Reply to the reviewer q9yF**
> > > > >
> > > > > Thank you very much for your thoughtful engagement during the review process and for your kind words. We greatly appreciate your time and effort in providing valuable feedback, which has helped us improve the clarity of our work.
> > > > >
> > > > > We are grateful for your positive reassessment and the score adjustment, and we look forward to further contributing to the research community.

---

### Official Review · Reviewer_J495 · 2025-07-01

**Clarity:** 3
**Significance:** 3
**Originality:** 3
**Rating:** 4
**Confidence:** 3

**Summary:**

This paper aims at solving a more realistic few-shot learning challenge called Dual Support-Query Shift (DSQS), which accounts for both inter-set and intra-set distribution shifts. To address this, the authors propose DUal ALignment framework (DUAL) that combines a pixel-level repairer and an adversarial generator to obtain clean, robust features, followed by a two-stage optimal transport alignment to reduce distribution mismatches. The method is theoretically justified and empirically validated, achieving strong performance gains over baselines across multiple benchmarks.

**Questions:**

1, Can the authors clarify whether the shifts used to train the repairer R are the same as those applied during DSQS testing, or whether there is any empirical analysis on how the type of training shifts affects performance?

2, Although Table 5 reports the inference cost of the method, could you provide a comparison with baseline approaches to clarify how the added modules affect efficiency relative to existing methods?

3, Could you provide a comparison of the parameter number or computational cost of your method (including repairer, generator, and dual OT) against baseline approaches to clarify whether the added components significantly increase model complexity?

**Ethical Concerns:**

["NO or VERY MINOR ethics concerns only"]

**Final Justification:**

The authors have addressed my main concerns in the rebuttal. I am satisfied with their responses and will keep my original score.

**Limitations:**

yes

**Quality:**

3

**Strengths And Weaknesses:**

Strengths:

1.The paper provides solid theoretical analysis to support its design choices, including how the repairer and adversarial generator reduce intra-set noise and expand class margins, and how the dual-stage optimal transport improves distribution alignment and lowers classification risk. This adds depth and clarity to the proposed framework.

2.The method demonstrates significant and consistent improvements over various few-shot learning approaches across multiple benchmarks, including CIFAR-100, mini-ImageNet, and Tiered-ImageNet. The gains are especially pronounced under challenging distribution shift settings, validating the practical effectiveness of the proposed solution.

Weaknesses:

1.In line 131, the variables $w$ and $\pi$ used in the optimal transport formulation are not clearly defined in the main text. While readers familiar with OT may infer their meaning, it would improve clarity and self-containment if the paper explicitly defined these terms when first introduced.

2.The paper relies heavily on optimal transport (OT) but gives only a minimal explanation. A brief background on OT and clarification of the solution process would greatly improve clarity for readers unfamiliar with this concept.

---

> ### Author Rebuttal · Authors · 2025-07-30
>
> Thank you for your thoughtful feedback. We appreciate your suggestions and will address them to improve the clarity and accessibility of our work.
>
> **Q1:** *More descriptions of Optimal transportation.*
>
> **A1:**  Regarding the variables $w$ and $\pi$ in line 131, we agree that explicitly defining these terms in the main text would enhance clarity and ensure the paper is more self-contained. While readers familiar with optimal transport (OT) may infer their meanings, we will revise the manuscript to provide clear definitions and explanations when introducing these terms. In addition, we also acknowledge that the paper relies heavily on optimal transport (OT), and a more detailed explanation of its background and the solution process would aid readers unfamiliar with the concept. To address this, we will expand the related sections to include a concise yet comprehensive overview of OT, along with a step-by-step clarification of the solution process. This will ensure that the paper is more accessible to a broader audience.
>
> ---
> **Q2:** *Can the authors clarify whether the shifts used to train the repairer R are the same as those applied during DSQS testing, or whether there is any empirical analysis on how the type of training shifts affects performance?*
>
> **A2:**  Thank you for your insightful question. To clarify, we strictly separate the shift types used during training and testing to avoid any potential data leakage, adhering to the SQS protocols [1]. Specifically, all shifts used to meta-train the repairer R are entirely disjoint from those used in DSQS meta-testing. This ensures a fair and rigorous evaluation of our framework.
>
> ---
>
> **Q3:** *Although Table 5 reports the inference cost of the method, could you provide a comparison with baseline approaches to clarify how the added modules affect efficiency relative to existing methods?*
>
> **A3:** Thank you for raising this important point. To address your concern, we have conducted a comparison of inference costs between DUAL and baseline methods on Tiered-ImageNet, using the 8-target 1-Shot inference setup.
>
> As shown in Table R1-J495, it reveals that DUAL (ProtoNet) increases inference time by only **6%** compared to the vanilla ProtoNet. This level of overhead is comparable to MatchingNet (**7%**) and on par with other SQS baselines, such as TP (**6%**) and TransPropNet (**3%**). These results demonstrate that our method introduces only a minimal additional inference cost while achieving significant improvements in addressing the DSQS challenge.
>
> >Table R1-J495: Inference cost of DUAL.
> |    Method    | Average Latency (ms) | Relative Cost (%) |
> |:------------:|:--------------------:|:-----------------:|
> |   ProtoNet   |         5.80         |       100%        |
> | MatchingNet  |         6.21         |       107%        |
> |      TP      |         6.17         |       106%        |
> | TransPropNet |         5.97         |       103%        |
> |  **DUAL-P**  |       **6.14**       |     **106%**      |
>
>
> ---
>
>
> **Q3:** *Could you provide a comparison of the parameter number or computational cost of your method (including repairer, generator, and dual OT) against baseline approaches to clarify whether the added components significantly increase model complexity?*
>
> **A3:** Thank you for your thoughtful question. We appreciate the opportunity to elaborate on this aspect of our method. To address your concern, we provide a detailed breakdown of the parameter distribution, as shown in Table R2-J495.
>
> In total, DUAL introduces **~22 %** additional parameters compared to a parameter-free baseline such as ProtoNet or TP. Specifically, during inference, the repairer is implemented as a single shallow RedNet, while the dual OT is solved just once per episode using lightweight Sinkhorn iterations.
>
> This carefully designed architecture ensures that the added complexity remains modest, with minimal impact on runtime efficiency. As demonstrated by the inference latency results in Table 5, the overhead introduced is negligible. We believe this balance between enhanced performance and computational efficiency highlights the practicality and scalability of DUAL in addressing the challenges of DSQS.
>
> >Table R2-J495: Comparison of the parameter number of DUAL.
> |     Component      |   Params (M)    | Share of Total |
> |:------------------:|:---------------:|:--------------:|
> | Embedding backbone |      11.18      |      78 %      |
> |     Repairer R     |      3.13       |      21 %      |
> |    Generator G     |      0.07       |     0.5 %      |
> |   Dual‑OT plans    | 0 (solver only) |       –        |
>
>
> [1] Bridging Few-Shot Learning and Adaptation: New Challenges of Support-Query Shift

---

> > ### Comment · Area_Chair_Diw5 · 2025-08-06
> >
> > Dear Reviewer,
> >
> > The authors have provided their rebuttal with additional results. Please kindly reply to them as soon as possible before the discussion period ends.
> >
> > Thanks a lot.
> >
> > Best regards,
> >
> > AC

---

> > ### Comment · Reviewer_J495 · 2025-08-06
> >
> > Thank you for your detailed response. I appreciate the authors’ efforts in addressing my concerns. I am satisfied with the response and will keep my original score.

---

> ### Author Response · Authors · 2025-08-06
> **Reply to Reviewer J495**
>
> Thank you for taking the time to review our detailed response and for recognizing our efforts in addressing your concerns. We sincerely appreciate your thoughtful feedback and are glad to know that you are satisfied with our clarifications.
>
> While we fully respect your decision to maintain your original score, we hope that our rebuttal has effectively addressed all of your concerns. **If so, we kindly ask you to consider raising your score, as your support would greatly contribute to the recognition and dissemination of our work.** That said, we completely understand if you choose to keep your original evaluation, and we remain deeply grateful for your valuable feedback and guidance.
>
> Thank you once again for helping us improve the quality of our work.

---

### Official Review · Reviewer_VWnr · 2025-07-01

**Clarity:** 3
**Significance:** 2
**Originality:** 2
**Rating:** 4
**Confidence:** 4

**Summary:**

This paper aims to solve the problem of few-shot learning (FSL), where the goal is to achieve good generalization on query set, given a small support set. The authors claim to solve the issue of two shifts in support and query datasets, inter-set shifts and intra-set shifts. They propose dual alignment framework for feature embeddings to alleviate these issues, and they implement (1) a repairer module for inter-set and intra-set distortion rectification, and (2) a generator module for simulating hard examples which encourages purturbation. They validate their method on well-known few-shot classification benchmarks, comparing their method to other algorithms.

**Questions:**

Please refer to the weaknesses section for questions. The authors need further validation for the novelty behind their contributions.

**Ethical Concerns:**

["NO or VERY MINOR ethics concerns only"]

**Final Justification:**

Most of the issues raised by the reviewers were addressed by the authors to some degree, and I am inclined to raise my rating of the paper.

**Limitations:**

Yes

**Quality:**

2

**Strengths And Weaknesses:**

- Strengths
  - Motivation behind few-shot learning and its issues are plausible, and the questions raised by the authors are appropriate. The authors' proposed approach is well-aligned to tackle these issues.
  - Authors show the performance of their approach on well-known few-shot learning benchmarks, comparing their method to relatively recent few-shot learning baselines with similar (prototypical, exemplar) structure.

- Weaknesses
  - Since there is a previous publication (PGADA [19]) with highly similar motivation and contributions, authors should clearly point out the differences and make their novelty more prominent.
    - Additional missing reference with similar approach: "Dual Adaptive Representation Alignment for Cross-Domain Few-Shot Learning", Zhao et al., TPAMI 2023.
	- "Cross-Domain Few-Shot Classification via Learned Feature-Wise Transformation", Tseng et al., ICLR 2020.
  - Since the main contribution of the paper is to deal with intra- and inter-set shifts for better generalization across few-shot tasks, experimental ablation for the cross-domain FSL scenario should be included. Although cross-domain FSL assumes domain shift between meta-training and meta-test tasks, its results should roughly quantify the generalizability of the proposed method.
  - The generator G samples a hard augmented image $x_p$ from $P$ which is Gaussian, and the Gaussian perturbation seems to be too simple for real-world visual domains. Since there are many augmentation methods (MixUp, CutMix, AugMix, etc.) available, why are these methods not used?

---

> ### Author Rebuttal · Authors · 2025-07-30
>
> Thank you for the reviewer’s comments.  In the following rebuttal, we aim to clarify the questions and have provided detailed responses to address each concern comprehensively.
>
> **Q1:** *Clarify the difference between DUAL and PGADA [19]*.
>
> **A1:** Thank you for your comments. While our motivation is similar to PGADA, our work makes three significant contributions that distinguish it from PGADA:
> - Dual Support-Query Shift (DSQS) Challenge: In our framework, we are the first to identify and propose the Dual Support-Query Shift (DSQS) challenge.
> - Theoretical Support: We provide theoretical evidence demonstrating that both inter- and intra-distribution shifts can misguide the domain alignment process, which has not been addressed in PGADA.
> - Novel Framework: We introduce a new framework, DUAL, which incorporates a dual adversarial training scheme and a dual optimal transportation scheme to effectively address the DSQS challenge.
> These contributions highlight the originality and practical value of our work compared to PGADA. **For more details, please refer to our Introduction, Page 2, Line 73-83.**
>
> ---
> **Q2:** *Clarify the difference and experimental ablation for the cross-domain FSL (CD-FSL).*
>
> **A2:** Thank you for your comments. We have clarified the differences between DSQS and CD-FSL in **footnote 2 on page 2**. While CD-FSL assumes a domain shift between meta-training and meta-testing tasks, a key distinction is that it typically considers the shift to originate from a single domain [1-3]. In contrast, DSQS addresses a more complex and realistic scenario where shifts can arise from multiple domains. This means that instances within either the support set or the query set may belong to different domains, introducing a higher level of diversity and complexity. As a result, DSQS represents a fundamentally different and more challenging problem compared to CD-FSL.  To further strengthen our discussion, we will update the related work section to include discussions on [TPAMI 23] and [ICLR 20] in the context of CD-FSL.
>
> [1] A Closer Look at the CLS Token for Cross-Domain Few-Shot Learning. NeurIPS 2024.
>
> [2] Cross-Domain Cross-Set Few-Shot Learning via Learning Compact and Aligned Representations ECCV 2022.
>
> [3] Understanding Cross-Domain Few-Shot Learning Based on Domain Similarity and Few-Shot Difficulty NeurIPS 2022.
>
> ---
>
>
> **Q3:** *Why not use more complex augmentation methods such as MixUp, CutMix, AugMix.*
>
> **A3:** Thank you for your comments. We agree that more complex augmentation methods, such as MixUp, CutMix, or AugMix, could be incorporated into our framework, as they can serve as plug-and-play components to replace the generator $G$. However, it is important to emphasize that the primary contribution of DUAL is the introduction of a novel framework specifically designed to address the DSQS challenge, rather than the development of advanced, high-performance data augmentation techniques. Our focus is on providing a flexible and effective solution for mitigating DSQS, which could be further enhanced by integrating such augmentation methods if desired.

---

> > ### Comment · Area_Chair_Diw5 · 2025-08-06
> >
> > Dear Reviewer,
> >
> > The authors have provided their rebuttal. Please kindly reply to them as soon as possible before the discussion period ends.
> >
> > Thanks a lot.
> >
> > Best regards,
> >
> > AC

---

> > ### Comment · Reviewer_VWnr · 2025-08-06
> >
> > - Regarding **Q1**, even with the explanations provided by the author, the proposed method still lack fundamental differences compared to PGADA, in both aspects of motivation and approach. I acknowledge that the authors are the first to address the intra-set shifts, but its implementation involves only class-wise centroids $\bar{S}$ in equation (10), resulting in very high text overlap between this paper and PGADA. The authors should provide additional points stating fundamental differences between both works, rather than minor differences in the implementations.
> >
> > - Regarding **Q2**, although I acknowledge the differences between CD-FSL task and the motivation of DSQS, the experimental validations in this paper only involve homogeneous tasks that are sampled from the identical datasets, which is far less challenging compared to CD-FSL. CD-FSL involves heterogeneous support and query tasks sampled from different datasets, and is a more realistic setting compared to conventional FSL settings, and validation under this setting will further strengthen the practicality of the proposed method.

---

> > > ### Author Response · Authors · 2025-08-07
> > > **Response to Reviewer VWnr - (Part 2/2)**
> > >
> > > **2. Addressing the Scope of Experimental Validation**
> > >
> > > We sincerely thank the reviewer for their insightful suggestion regarding the scope of our experimental validations. However, firstly, we would like to respectfully clarify our perspective on the statement:
> > > > CD-FSL involves heterogeneous support and query tasks sampled from different datasets.
> > >
> > > Based on our experience with CD-FSL, particularly as outlined in the benchmarking paper on Cross-Domain Few-Shot Learning (CD-FSL) [1], we would like to clarify a potential misunderstanding. **CD-FSL, as defined in the literature, does not exhibit heterogeneity between the support set and the query set. Instead, both the support set and query set are sampled from the same dataset to maintain a shared label space.** The heterogeneity arises primarily between the training and testing phases, where the datasets differ. We hope this clarification helps refine the understanding of the CD-FSL setting.
> > >
> > > The key reason for this design lies in the requirement that the label space in N-way K-shot tasks must be shared. In other words, if the instances in the support set and query set were to come from different datasets, it would be challenging to ensure that the selected categories align correctly between them.
> > >
> > > Then, we agree with your point that:
> > >
> > > > support and query tasks sampled from different distribution (maybe from different datasets) is a more realistic setting compared to the convetional FSL setting.
> > >
> > > To address this, we proposed DSQS to simulate such a setting while ensuring the shared label space required for N-way K-shot tasks is maintained.
> > >
> > > Furthermore, our experiments on the meta-Dataset provide insights into both CD-FSL and DSQS. For example, as shown in Table 3, by focusing solely on DSQS, the results on ImageNet-1K demonstrate that DUAL performs well compared to selected baselines. Similarly, when extending the analysis to CD-FSL + DSQS (cross domain dual support-query shift FSL), we observe consistent performance improvements.
> > >
> > > For further understand how CD-FSL baselines perform in such scenarios. we selected two SOTAs, i.e., TSA (CVPR 2022) [2] and DIPA (CVPR 2024) [3], for a quick validation. Specifically, we applied our shift on the Meta-Dataset using ResNet-18 as the backbone for these methods. For DIPA, the backbone is ViT-S, while TSA and DUAL-P use ResNet-18 as the backbone.  For clarity:
> > > - **In-domain**: Trained and evaluated on ImageNet-1K.
> > > - **Out-of-domain**: Trained on ImageNet-1K and evaluated on other datasets (such as Aircraft, Describable Textures, and MSCOCO, etc.).
> > >
> > >
> > > > TableR1-VWnr: Evaluation with Cross-Domain Few-Shot Learning (CD-FSL) Baselines
> > >
> > > | Method | In-Domain (w/o DSQS) | Out-of-Domain (Avg) (w/o DSQS) | In-Domain (w/ DSQS) | Out-of-Domain (Avg) (w/ DSQS) |
> > > |--------|:--------------------:|:-----------------------------:|:------------------:|:-----------------------------:|
> > > | TSA    |        59.50         |             80.40             |        18.33       |             29.34            |
> > > | DIPA   |     **70.90**        |          **82.60**            |        19.22       |             26.54            |
> > > | DUAL-P |        63.27         |             68.89             |     **25.66**      |          **39.07**           |
> > >
> > > From the results, we observe that the performance of the methods depends heavily on where the difficulty is introduced. Specifically, CD-FSL methods do not account for the support-query shift, which negatively impacts their performance. However, DUAL-P is not explicitly designed to address cross-domain issues. Its relatively strong performance might stems from the robustness of the model itself.
> > >
> > > We acknowledge that the CD-FSL and DSQS tasks both represent a more realistic and challenging setting compared to conventional FSL.  While our current experiments focus on homogeneous tasks sampled from identical datasets, we agree that extending the validation to the CD-FSL setting would further strengthen the practicality and generalizability of our proposed method.
> > >
> > > We deeply appreciate this valuable feedback which can help us to better demonstrate the adaptability and robustness of our framework under more diverse and realistic conditions.
> > >
> > > [1] A broader study of cross-domain few-shot learning. ECCV 2020.
> > >
> > > [2] Cross-domain few-shot learning with task-specific adapters CVPR 2022.
> > >
> > > [3] Discriminative Sample-Guided and Parameter-Efficient Feature Space Adaptation for Cross-domain Few-Shot Learning  CVPR 2024.

---

> ### Author Response · Authors · 2025-08-07
> **Response to Reviewer VWnr - (Part 1/2)**
>
> **1. Technical Contributions of the DUAL Framework Compared to PGADA**
>
> Firstly, our goal is to obtain clean and robust features to improve the performance of optimal transport (OT) in aligning distributions. We fully agree that OT performs best when features are well-represented and robust. Building on this idea, we would like to highlight the key technical contributions of the DUAL framework that distinguish it from PGADA, as outlined below:
>
> **1.1 Comparison with PGADA (Model Design Perspective):**: Our approach introduces two additional components within a two-stage framework:
>
> - **Repairer Network ($R$)**: We incorporate the repairer network during inference to obtain clean features, significantly enhancing the robustness of feature representation.
> - **Class-wise Centroids for Alignment**: As noted, Equation (10) utilizes class-wise centroids for alignment. While this may not seem like a fundamental difference at first glance, it plays an integral role in our broader technical design, which is specifically tailored to address intra-set shifts.
>
> **1.2 Comparison with PGADA (Broader Contributions)**:
> - Beyond these differences in framework design, one of our primary contributions is the introduction of a new challenge (DSQS) to the field, offering deeper insights and advancing research in this area.
> - In addition, we have conducted extensive theoretical analyses, including Proposition 1, Proposition 2, Lemma 1, Lemma 2, and Theorem 1 (Pages 6-8).
> - Furthermore, our work introduces new experimental evaluations, including comprehensive large-scale experiments, such as the meta-dataset described on Page 18. These aspects, which were not addressed in PGADA, highlight the novelty and practical significance of our contributions.
>
> We believe these distinctions and contributions demonstrate the unique value and impact of our approach compared to PGADA.

---

### Official Review · Reviewer_H8EC · 2025-07-03

**Clarity:** 2
**Significance:** 2
**Originality:** 3
**Rating:** 4
**Confidence:** 3

**Summary:**

The manuscript introduces DUAL, which tackles the challenging Dual Support-Query Shift (covering both inter-set and intra-set shifts) in few-shot learning. It extracts clean features via a robust embedding function combined with a pixel-level repairer, and then employs a dual-regularized optimal-transport approach to align the distributions of instances in the support and query sets. Extensive experiments on multiple datasets verify the effectiveness of the proposed method.

**Questions:**

1. Because the paper’s main contribution is to mitigate inter-set and intra-set shifts, please provide visualizations that clearly show which step alleviates inter-set shifts and which operation resolves intra-set shifts; this would better substantiate the central claim.

2. The section on the two-stage OT approach is under-described: many implementation details are missing, and the experiments do not show that the two-stage method offers any advantage over a single-stage alternative.

3. Methodological details remain under-specified. Prior work largely focuses on the support-query shift (inter-set shift); because this paper additionally introduces intra-set shifts, more fine-grained methodological details are needed to clarify exactly how intra-set shifts are mitigated.

4. Some experimental aspects remain unclear: 1) How stable and robust are the hyper-parameters λ and β?.  2) In Appendix B (Details of Experiments), why were those particular network structures (e.g., REDNET) selected?

5. Most core experiments use CIFAR-100, yet the “Computation Overhead” paragraph states, “For training, CIFAR-10 is the least computationally demanding…”. Is this a typo? The t-SNE visualization is also labeled as CIFAR-10.

**Ethical Concerns:**

["NO or VERY MINOR ethics concerns only"]

**Final Justification:**

Most of my concerns have been resolved, and the authors have promised to address the remaining issues in the revised manuscript. As a result, I have raised my score to 4.

**Limitations:**

Yes

**Quality:**

3

**Strengths And Weaknesses:**

Strengths:
1. Extensive experiments validate the method on multiple datasets, with comprehensive ablation studies.

2. Introducing the dual support-query shift scenario offers a certain degree of novelty.

3. Clear writing flow. The overall narrative of the paper is smooth and well-structured.

Weaknesses：
1. Some essential experiments have not yet been reported. For instance, the paper targets both inter-set and intra-set shifts, certain experiments and visualizations do not convincingly isolate which operations address inter-set shifts and which tackle intra-set shifts.

2. Methodological details remain under-specified. Prior work largely focuses on the support-query shift (inter-set shift); because this paper additionally introduces intra-set shifts, more fine-grained methodological details are needed to clarify exactly how intra-set shifts are mitigated.

3. Presentation quality still needs improvement. The manuscript contains some minor errors throughout.

---

> ### Author Rebuttal · Authors · 2025-07-30
>
> Thank you for the reviewer’s comments. We have noticed that many of the concerns raised stem from misunderstandings of our work. In the following rebuttal, we have addressed them point by point.
>
>
> **Q1 & Q2 & Q3:** *Thank you for your comment. We summary the comment and listed as the following 2 question. (1) Missing Methodological and implementation details and (2)  Effects and Visualizations of detailed relieving inter- and intra distribution shift.*
>
> **A1 & A2 & A3:**
>
> * (1) Missing Methodological and implementation details.
>     * (1-1 Methodology Details ) We acknowledge that our main paper provided a less detailed description of the introduction to dual optimal transportation. While this concept may be familiar to readers with prior knowledge (as noted by Reviewer J495 and Reviewer q9yF), we plan to include a more comprehensive explanation in the revised version to improve readability and understanding for a broader audience.
>     *  (1-2 Implementation Details) In the implementation details, we will provide additional illustrations of optimal transportation and have also provided our reproducible code to further clarify our paper.
>
> * (2) Effects and Visualizations of detailed relieving inter- and intra distribution shift.
>    * (2-1 Detailed Effect) To validate the effectiveness of our framework, we conducted 8 experiments, including 3 ablation studies and 5 in-depth analyses, as shown in Table 2. Additionally, to further evaluate the impact of alleviating inter- and intra-distribution shifts in detail, we provide an experiment under the 5-way 5-shot setting, specifically focusing on their removal, **as shown in Table R1**.
>    * (2-2 Visualizations) We have provided t-SNE visualizations after adopting our framework in Fig. 4. To further illustrate the effect of alleviating inter- and intra-distribution shifts, we use cosine similarity to measure the distance between embeddings after applying our method. Specifically, we select several samples from the CIFAR-100 dataset to construct support and query sets from different classes and extract their embeddings to compare the cosine similarity of these samples. The key difference in this comparison is whether or not dual optimal transportation (dual OT) is applied.  **As shown in Table R2**, we observe that the distances are closer when dual OT is used, demonstrating that our approach effectively reduces the distribution shift. We believe these experiments provide comprehensive evidence supporting our conclusions.
>
> > Table R1 Impact of Removing Single-Stage OT
> |          Method          |  CIFAR-100  | mini-ImageNet | Tiered-ImageNet |
> |:------------------------:|:-----------:|:-------------:|:---------------:|
> |      w./o. dual OT       | 50.76+-0.38 |  65.61+-0.39  |   44.46+-0.38   |
> | w./o. intra-OT ($\pi'$)  | 52.12+-0.37 |  66.10+-0.38  |   45.82+-0.39   |
> | w./o. inter-OT ($\pi^*$) | 52.21+-0.31 |  66.75+-0.40  |   46.20+-0.40   |
> |           DUAL           | 54.47+-0.40 |  67.83+-0.40  |   47.81+-0.41   |
>
> > Table R2 Cosine Similarity of Visualizations
> |  Method   | Orginal | Add Shifts | Adopting intra-OT | Adopting inter-OT |
> |:---------:|:-------:|:----------:|:-----------------:|:-----------------:|
> | CIFAR-100 | 0.9424  |   0.7289   |      0.8549       |      0.8728       |
>
>
> ---
> **Q4:** *Unclear experimental aspects, e.g., hyper-parameters and REDNET structures.*
>
> **A4:** We employed a grid search approach to explore the stability and robustness of our hyperparameters, as detailed in **Table 7, Appendix C.6**. We have tested several combinations of $\lambda \in (0.1, 0.5, 0.9)$ and $\beta  \in (0.1, 0.5, 0.9)$. Our findings indicate that the best performance in our setting is achieved when $\lambda=0.5$ and $\beta=0.5$. Regarding the REDNET structure, its use is primarily due to its widespread adoption as a robust image restoration  architecture [1-3]. Additionally, we want to emphasize the flexibility of our DUAL framework. Notably, other approaches, such as GANs or vision transformers, can also be substituted for REDNET while maintaining the compatibility and effectiveness of our framework.
>
> [1] SAVSR: Arbitrary-Scale Video Super-Resolution via a Learned Scale-Adaptive Network. (AAAI 2024).
>
> [2] FADE: A Task-Agnostic Upsampling Operator for Encoder–Decoder Architectures.(IJCV 2025).
>
> [3] Fast and accurate single image super-resolution via information distillation network. (CVPR 2018)
>
> ---
> **Q5:** *Typos in the Appendix.*
>
> **A5:** Thank you for pointing this out. All experiments are conducted on CIFAR-100. We will thoroughly review the entire paper, including the Appendix, and make the necessary revisions.

---

> > ### Comment · Area_Chair_Diw5 · 2025-08-06
> >
> > Dear Reviewer,
> >
> > The authors have provided their rebuttal with additional results. Please kindly reply to them as soon as possible before the discussion period ends.
> >
> > Thanks a lot.
> >
> > Best regards,
> > AC

---

> > ### Comment · Reviewer_H8EC · 2025-08-06
> >
> > Thank you for the rebuttal. It has addressed most of the concerns raised. My only remaining concern is that the paper targets both inter-set and intra-set shifts. To make the proposed method more convincing, it would be more persuasive to include visualizations in the manuscript that clearly demonstrate the results when each type of shift is applied separately.

---

> ### Author Response · Authors · 2025-08-06
> **Reply Reviewer H8EC**
>
> Thank you very much for your thoughtful feedback and for recognizing that our rebuttal has addressed most of your concerns. We sincerely appreciate your efforts in reviewing our work.
>
> Regarding your suggestion to include visualizations demonstrating the results for inter-set and intra-set shifts separately, we completely agree that such visualizations would make the paper more persuasive. In the final version of the manuscript, we are committed to incorporating these visualizations to clearly illustrate the effectiveness of our method under these specific conditions.
>
> Additionally, some of our experimental results (Table R2) already demonstrate that our approach effectively alleviates the challenges posed by both types of shifts. Furthermore, we have included two additional datasets for further validation **(Table R3)**. We believe that incorporating visualizations will further strengthen these findings and enhance the clarity and impact of our contributions.
>
> > Table R3 More results on cosine similarity of visualizations.
> |     Method      | Orginal | Add Shifts | Adopting intra-OT | Adopting inter-OT |
> |:---------------:|:-------:|:----------:|:-----------------:|:-----------------:|
> |    CIFAR-100    | 0.9424  |   0.7289   |      0.8549       |      0.8728       |
> |  mini-ImageNet  | 0.9639  |   0.7698   |      0.8926       |      0.9128       |
> | Tiered-ImageNet | 0.8739  |   0.6827   |      0.7234       |      0.8099       |
>
>
> We hope you can understand that due to the limitations of OpenReview, we are currently unable to include visualizations directly in the rebuttal. However, we will ensure that they are comprehensively added to the final version of the paper.
>
> We hope that our rebuttal has effectively addressed all of your concerns. **If so, we kindly ask you to consider raising your score**, as your support would greatly contribute to the recognition and dissemination of our work.
> Your constructive feedback has been invaluable in improving the quality of our work, and we deeply appreciate your support.
>
> Thank you again for your time and understanding.

---

### Note · Authors · 2025-08-11

Dear AC Diw5 and all Reviewers,

We sincerely appreciate your time and effort to review our work.

Firstly, we would like to extend our gratitude to Reviewers q9yF and J495 for recognizing the value of our work (with one awarding a score of 4/6 and the other indicating an intention to increase their score from 4 to 5). In particular, we are deeply grateful to **Reviewer q9yF** for providing excellent suggestions that significantly enhanced the quality of our paper, especially in terms of clarity and readability.

---

Secondly, we are pleased to have addressed most of **Reviewer H8EC's** concerns. In our final response, we included a detailed analysis of the visualizations (the remaining issue), which we have committed to incorporating into the revised manuscript. We firmly believe that our response has resolved the remaining issue and hope this will encourage them to **reconsider and raise** their score.

---

Lastly, regarding **Reviewer VWnr**, we noticed that their main concerns focus on two key issues (out of the three they raised).

1. It seems this reviewer places significant emphasis on the technical contributions of our work and its differentiation from PGADA. However, we would like to clarify that the technical contributions represent only a small part of DUAL. In addition to this, DUAL identifies a ```novel challenge, introduces new theoretical insights, and provides innovative experiments, none of which are present in PGADA.```
We hope this highlights the broader scope of our contributions.

2. We realize that there might be a misunderstanding between us and Reviewer VWnr regarding the concept of CD-FSL. From our perspective, it seems that Reviewer VWnr interprets CD-FSL as an enhanced version of DSQS, where the support set and query set are from different domains (datasets), aligning more closely with real-world scenarios. However, based on our experience with CD-FSL, it does NOT achieve this goal at all. CD-FSL typically ensures that the ```training and testing datasets come from different domains (datasets) but requires the support set and query set to share the same label space```, i.e., from the same datasets during meta-training or meta-testing.

We deeply appreciate Reviewer VWnr’s acknowledgment of the authenticity of the problem we address. And we believe that the quick evaluation (TableR1-VWnr) in the response has effectively clarified the difference between DSQS and CD-FSL.

We hope the information helps the AC in making a decision.

---

### Decision · Program_Chairs · 2025-09-17

**Decision:**

Accept (poster)

**Comment:**

This paper proposes DUAL, a framework for few-shot learning under Dual Support-Query Shift (DSQS), integrating inter-set and intra-set distribution shifts. The idea is well-motivated and supported by both theoretical analysis and extensive experiments. The paper presents technically sound work with clear contributions to the few-shot learning domain. Reviewers agree that the problem formulation is novel and valuable, and the framework shows consistent gains over strong baselines.

While initially there were concerns about clarity, relation to PGADA, and experimental scope, the authors have demonstrated commitment to addressing these issues comprehensively. These were convincing to most reviewers, who raised or maintained positive borderline scores. Therefore, I would recommend acceptance of the paper. For camera-ready, I encourage the authors to:

1. Incorporate all promised writing improvements for clarity
2. Add the visualization analyses discussed in the rebuttal
3. Include the cross-domain FSL discussion and results